# Exploration of crystal chemical space using text-guided generative artificial intelligence

Hyunsoo Park ✉, Anthony Onwuli & Aron Walsh ✉

The vastness of chemical space presents a long-standing challenge for the exploration of new compounds with pre-determined properties. In materials science, crystal structure prediction has become a mature tool for mapping from composition to structure based on global optimisation techniques. Generative artificial intelligence now offers the means to efficiently navigate larger regions of crystal chemical space informed by structure-property datasets of materials. Here, we introduce a model, named Chemeleon, designed to generate chemical compositions and crystal structures by learning from both textual descriptions and three-dimensional structural data. The model employs denoising diffusion techniques for compound generation using textual inputs aligned with structural data via cross-modal contrastive learning. The potential of this approach is demonstrated for multi-component compound generation, including the Zn-Ti-O ternary space, and the prediction of stable phases in the Li-P-S-Cl quaternary space of relevance to solid-state batteries.

The true power of artificial intelligence (AI) for chemical research is in addressing challenges that are difficult or impossible to solve using conventional methods. Already, AI techniques are being tailored to identify patterns in chemical datasets[1], predict reaction outcomes[2], and accelerate discovery and optimisation cycles[3]. The efficient navigation of chemical space, exploring or targeting relevant regions in the high-dimensional composition-structure-property landscape, remains one of the longstanding goals in computational chemistry.

Computational materials exploration has expanded to incorporate both in-depth studies of individual systems[4] and high-throughput screening[5] approaches, each playing a complementary role in advancing the field. While high-throughput methods have significantly broadened the scope of materials discovery[6], in-depth studies remain essential for uncovering fundamental mechanisms and validating computational predictions. However, as these search spaces grow, so does the complexity of identifying regions of interest where chemical compositions, crystal structures, and physical properties align to create materials with desirable characteristics.

Navigating the vast pool of possible chemical compositions and molecular/crystal structures presents a significant challenge, akin to exploring a multidimensional surface, one step at a time[7]. While databases of known structures and properties provide a valuable

foundation, the search space is too expansive for exhaustive exploration using traditional methods alone. Chemical heuristics, grounded in empirical knowledge and informed by data, can play a crucial role in guiding this search to relevant regions, and in defining safe pathways to follow[8]. Accelerated random structure searching[9] and approximate probe structures[10] have also been used to identify fertile regions of chemical space

Generative AI now offers a complementary solution to these challenges, providing tools for data-driven optimisation and discovery[11]. By learning from both existing datasets and theoretical models, generative AI can be used to explore uncharted regions by drawing from the distributions of known materials to propose new candidate compounds with targeted properties. Early models based on generative adversarial networks (e.g., CrystalGAN[12]), and variational autoencoders (e.g., CDVAE[13]), are now being superseded by generative diffusion models (e.g., DiffCSP[14] and MatterGen[15]). These can be used as a tool for crystal structure prediction (mapping from chemical formula as input to candidate crystal structures as output) or the compound generation can be conditioned to achieve target property values in an inverse materials design workflow (mapping from property as input to candidate chemical formulas and crystal structures as output).

Department of Materials, Imperial College London, London, UK. ✉e-mail: h.park@imperial.ac.uk; a.walsh@imperial.ac.uk

In this work, we introduce a generative materials model, Chemeleon, based on denoising diffusion that learns from textual descriptions alongside three-dimensional structural data to sample chemical compositions and crystal structures. By incorporating textual descriptions into the training process, the model is better informed of the relationship between composition and structure. This approach is achieved by introducing cross-modal contrastive learning where embedding vectors from a text encoder are aligned with those from a crystal GNNs. The flexible nature of the model architecture can support the future integration of more sophisticated text descriptions and physical properties of materials.

## Results

To bridge the gap between textual descriptions and crystal structure generation, we have formulated a generative model trained using information from both types of data, as illustrated in Fig. 1a. The first component is a text encoder pre-trained via contrastive learning to align text embedding vectors from a text encoder with graph embedding vectors from equivariant graph neural networks (GNNs). The second component is a classifier-free guidance denoising diffusion model for composition and structure generation, which is iteratively trained to predict the temporal evolution of noise by incorporating the text embeddings obtained from the pre-trained text encoder.

### Contrastive learning with crystal structures: Crystal CLIP

Creating numerical representations of text (i.e., text embeddings) that capture information about crystal structures is essential for generative models based on textual inputs. Mat2Vec[16] is a pioneering approach to construct text embedding in the domain of materials science, by training a Word2Vec[17] model on an extensive corpus of materials science literature. This method produces dense embedding vectors that facilitate the understanding of word meanings by situating similar words close to each other. Further enhancement in text embedding quality has been achieved through transformer-based methods[18] such as bidirectional encoder representations from transformers (BERT)[19] which capture the contextual relationships within sentences by pre-training on tasks such as masked language modelling. Several Transformer encoder models including SciBERT[20], MatSciBERT[21], and MatBERT[22], have been developed to train BERT with an enormous corpus of materials science literature. Notably, MatTPUSciBERT[23], which is built on the SciBERT and has been pre-trained with approximately 700,000 articles in materials science, has demonstrated high

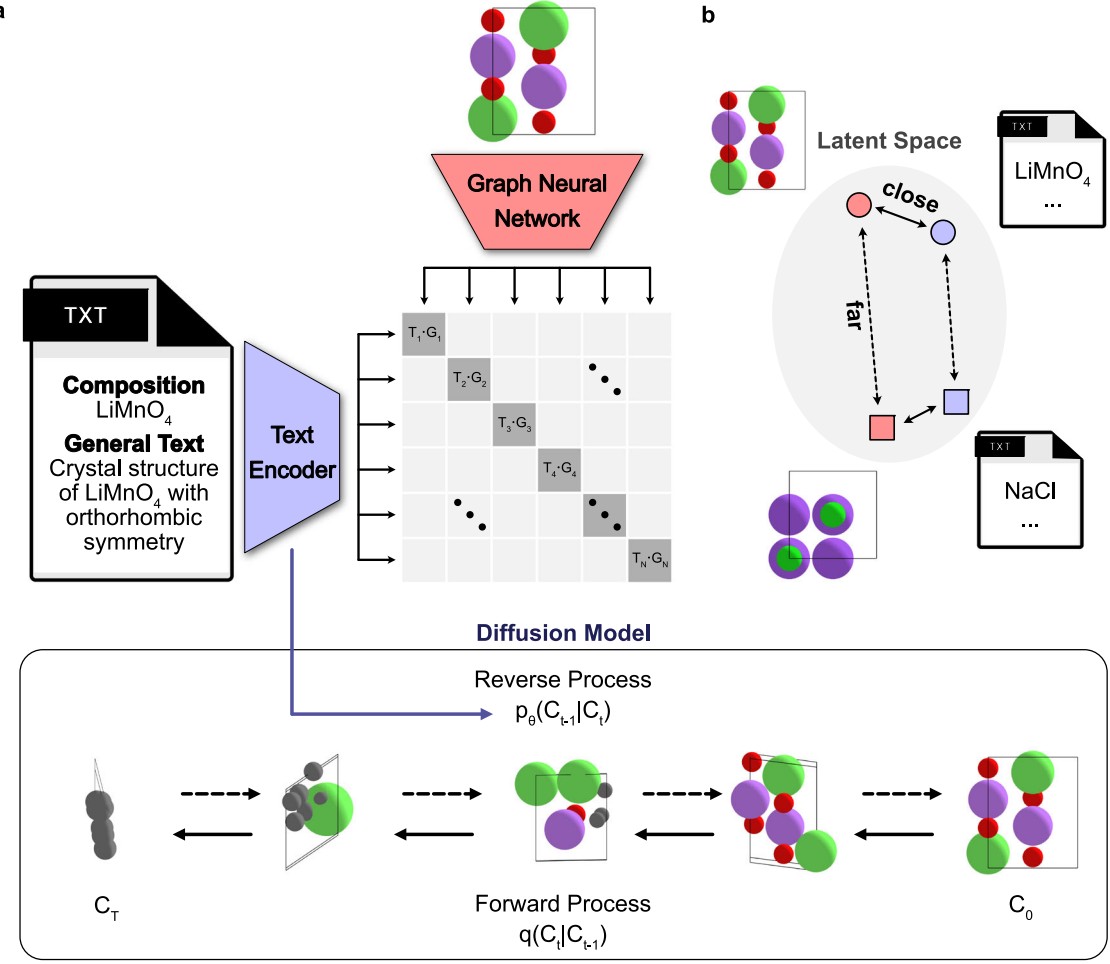

**Fig. 1 | Illustration of the cross-modal contrastive learning and generative diffusion approach implemented in Chemeleon. a** The text-guided denoising diffusion model comprises two key components: (1) Crystal CLIP (Contrastive Language-Image Pretraining), a text encoder pre-trained through contrastive learning to align text embeddings with graph neural network (GNN) embeddings derived from crystal structures, and (2) a classifier-free diffusion model, which iteratively predicts noise at each time step while integrating text embeddings from the pre-trained Crystal CLIP. In this framework, $q$ denotes the forward diffusion process (posterior) progressively adding noise to the crystal structures, while $p_\theta$ represents the reverse diffusion process (learned approximation of the posterior) aimed at generating crystal structures. $C_t$ refers to crystal structures at time step, $t$. **b** Illustration of the contrastive learning objective in Crystal CLIP, where positive pairs, which consist of text and graph embeddings from the same crystal structures, are brought closer together in the latent space, while negative pairs are pushed further apart.

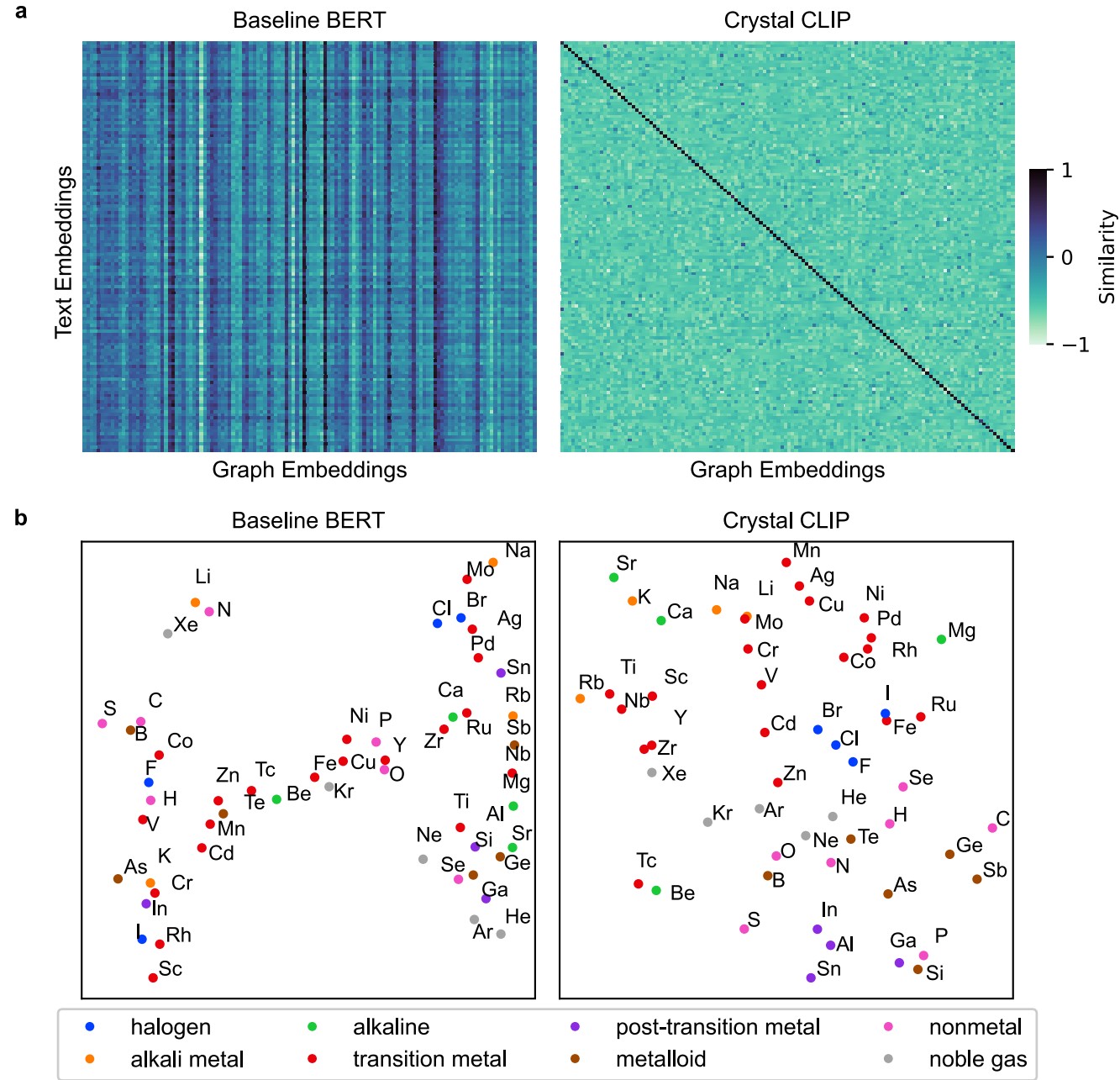

**Fig. 2 | Contrastive learning of text and crystal graphs. a** Heatmaps of cosine similarity between text embeddings from text encoders and graph embeddings from graph neural networks (GNNs). The Baseline BERT (Bidirectional Encoder Representations from Transformers) model refers to MatSciTPUBERT[23]. Crystal CLIP (Contrastive Language-Image Pretraining) denotes a text encoder trained using contrastive learning to align with graph embeddings. Values are plotted for 128 randomly sampled unit cells, forming a 128 × 128 matrix. Diagonal elements represent positive pairs, while off-diagonal elements represent negative pairs. **b** A t-SNE (t-distributed stochastic neighbor) visualization of element embeddings generated by the text encoders, using element symbols as the textual input.

performance in various tasks. However, such models may still exhibit limitations in their capacity to comprehend structure information in three-dimensional space and serve as effective text encoders for crystal structure generation.

To bridge the gap between text embeddings and accurate crystal structure representation, we have developed a cross-modal contrastive learning framework[24], named Crystal CLIP. This framework facilitates aligning text embeddings from text encoders with embeddings from other modalities. Demonstrations by various generative models of text to other modality, such as Imagen[25] and Dall-e[26], which incorporate text encoders pre-trained through contrastive learning with image embedding vectors, have shown enhanced performance

compared to models trained solely on textual data. For the specific task of capturing crystal structure information, Crystal CLIP is based on a Transformer-based encoder model that is trained through contrastive learning to align text embedding vectors with graph embeddings derived from GNNs.

Figure 2 illustrates the results of the contrastive learning of Crystal CLIP, which is based on MatTPUSciBERT (hereinafter referred to as Baseline BERT). The primary goal of this contrastive learning is to enhance the alignment between text and graph embeddings in the latent space. Specifically, the training objective is to maximize the cosine similarity for positive pairs, which consist of graph embeddings from GNNs and their corresponding textual descriptions, both derived

**Table 1 | Model evaluation on structural metrics**

| Model type | Textual description style | Text encoder | Validity | Uniqueness | Structure matching | Metastability |
|---|---|---|---|---|---|---|
| Composition | "Li1 Mn1 O4" | Baseline BERT | 0.99 | 0.94 | 0.13 | 0.22 |
| | | Crystal CLIP | 0.99 | 0.90 | 0.20 | 0.25 |
| Formatted text | "Composition: Li Mn O4, crystal system: orthorhombic" | Baseline BERT | 0.99 | 0.97 | 0.09 | 0.21 |
| | | Crystal CLIP | 0.98 | 0.92 | 0.17 | 0.19 |
| General text | "Crystal structure of LiMnO4 with orthorhombic symmetry" | Baseline BERT | 0.99 | 0.97 | 0.06 | 0.23 |
| | | Crystal CLIP | 0.99 | 0.90 | 0.20 | 0.25 |

Evaluation of the crystal structures generated by Chemeleon in terms of validity (structural parameters), uniqueness (between generated structures), structure matching (ground truth present), and metastability (energy threshold) across a batch of 20 samples.

from the same crystal structure. Simultaneously, the process minimizes the cosine similarity for negative pairs, which are not derived from the same crystal structure. This methodology ensures that positive pairs are positioned closely, and negative pairs are placed distantly in the embedding space, as illustrated in Fig. 1b.

Figure 2a presents heatmaps of cosine similarities for 128 randomly selected pairs of crystal structures and their textual descriptions which are reduced compositions of the crystal structures. The results for different types of textual descriptions are provided in Supplementary Informations S1 and 2. Within the heatmap, the diagonal elements, which represent positive pairs, show high cosine similarity, approaching 1. In contrast, the off-diagonal elements, representing negative pairs, display significantly lower cosine similarity, approaching −1. On the other hand, the Baseline BERT model fails to distinguish between positive and negative pairs, highlighting the utility of contrastive learning in Crystal CLIP.

To evaluate the effectiveness of contrastive learning, element embedding vectors generated by the text encoders, when the text input is an element symbol, are visualized using t-SNE[27] for dimensionality reduction, as illustrated in Fig. 2b. These embedding vectors are obtained from the class token, the initial token in the BERT architecture. The element embeddings from Crystal CLIP reveal distinct clustering of elements into groups such as transition metals, halogens, and noble gases. In contrast, the Baseline BERT model does not display such distinct clustering.

**Generative diffusion model**

The second stage of our framework is a diffusion model for generating compounds, which are represented by a crystallographic unit cell consisting of lattice vectors, atom types, and atom coordinates. The diffusion model operates with forward and backward processes[28]. The forward process involves gradually adding random noise (typically Gaussian noise) to the individual representations of crystal structures over a series of steps, transitioning the crystal structure ($C_0$) into a completely random state ($C_T$). The denoising model, which is based on an equivariant GNN ensuring E(3) symmetry, predicts the noise that was added at each step in the forward process and iteratively removes it, thereby reconstructing the original data from pure noise. The details of the diffusion model implementation are provided in the Methods section.

Beyond stochastically generating compounds from the original data distribution, Chemeleon is capable of guiding generation with textual descriptions toward specific types of composition or structure. Initially, guiding diffusion models required additional models alongside the diffusion model to predict specific conditional properties from the random noise at each step[29]. However, classifier-free guidance[30] eliminate the need for additional networks by explicitly incorporating the label (or conditioning data) into the input of the denoising model during both training and inference phases. The classifier-free guidance is adopted for Chemeleon where the text

embedding vectors from Crystal CLIP are used as conditioning data, effectively guiding the text-to-crystal structure generation process.

**Evaluation of structure metrics**

To explore a broad and diverse chemical space, our models were trained with inorganic crystal structures from the Materials Project[31] containing 40 or fewer atoms in the primitive unit cell. This approach allows Chemeleon to capture diverse material properties and structural variations, rather than training with only simple unit cells. Moreover, the test set was split chronologically to assess whether the models could generate (unseen) future structures. Detailed information about the dataset is provided in the Methods section and a comparison between chronological and random splits is discussed in Supplementary Note S1.

The textual descriptions of the crystal structures involve both the chemical composition and crystal system. Composition defines the element identity and ratio, fundamental factors that influence the local coordination preferences and long-range connectivity. The crystal system defines the shape (symmetry) of the unit cell, treated as a case property in this study. Three types of textual descriptions are used, as summarized in Table 1, each with an example. For the composition model, the textual description consists solely of the reduced composition in alphabetical order. The formatted text form represents structured data format by incorporating both the composition and an additional property, the crystal system, separated by a comma. The general text form allows for a more diverse range of textual descriptions, generated by large language models (LLMs). The details of generating textual descriptions and examples of general text are provided in the Method section and Supplementary Fig. S3.

Table 1 contains four evaluation metrics on structural metrics for diffusion models trained with the Baseline BERT and Crystal CLIP as text encoders. The test set comprises 708 structures that have been registered since August 2018, ensuring no overlap with the training or validation sets. For each test entry, the models generated 20 structures based on the number of atoms and text prompts derived from the test set structures which are considered the ground truth structures as the dataset possess feasible energy above the convex hull.

The Validity metric measures the proportion of structurally valid samples, where validity is defined by the absence of atomic overlap and cell lengths shorter than 60 Å. The absence of atomic overlap is evaluated by measuring the minimum pairwise interatomic distance. Structures with a minimum pairwise distance greater than 0.5 Å are considered free of atomic overlap. Both models achieve a near-perfect validity rate of 98–99%, indicating their robustness in generating structurally feasible outputs.

The *Uniqueness* metric evaluates the diversity of the generated structures, specifically the proportion of unique structures among the 20 samples for each text prompt based on the StructureMatcher class in Pymatgen[32] library. The Baseline BERT model exhibits a slightly higher unique score compared to Crystal CLIP, likely due to Crystal CLIP's tendency to generate structures with compositions that closely

match the given text prompts. Despite this, both models maintain a high uniqueness rate, exceeding 90%, which demonstrates their capability to produce varied outputs.

The Structure Matching metric assesses whether the 20 sampled structures include the ground truth structure from the test set, which was used to generate the corresponding textual description. Given that the test set consists of future data and including more than 20 atoms, this metric is particularly challenging. Nonetheless, Crystal CLIP can generate 20 % of unseen ground truth structures within the text set even when the type of textual description is general text. Notably, it significantly outperforms Baseline BERT, indicating its superior ability to replicate the precise structure as described by the text.

The Metastability metric measures the proportion of 20 sampled structures have an energy within 0.1 eV/atom of the ground truth structure's energy, based on MACE-MP[33] which is a pre-trained machine learning force-field from the energies and trajectories of geometry optimisation in Materials Project. This counting occurs only when the sampled structures share the same composition as the ground truth structures, and it is done without performing geometric optimization to allow for more efficient evaluation. This metric indicates how many of the generated structures are energetically close to the ground truth, suggesting their potential stability before further optimization. Both text encoders, Baseline BERT and Crystal CLIP, perform similarly in this regard, and about 19–25% of the sampled structures exhibit energetically favourable configurations without further optimization, demonstrating the models' effectiveness in generating stable structures directly from the initial sampling.

Although our model is not solely focused on the CSP task but rather emphasizes alignment for text-guided generation, we also trained it on the MP-20 dataset to facilitate a direct comparison with existing CSP models (see Supplementary Table S1). Unlike models only for the CSP task, which focus solely on optimizing atomic coordinates and lattice matrices, Chemeleon incorporates composition as a text-based input. As a result, it achieves a 67.52% composition matching rate, a metric that DiffCSP and FlowMM[34] do not account for. For structure match rate, our evaluation considers only cases where composition is correctly predicted. This stricter criterion naturally results in a lower structure match rate for Chemeleon compared to DiffCSP and FlowMM, as these models do not impose composition constraints. In terms of root mean square error, Chemeleon underperforms DiffCSP and exhibits similar performance with FlowMM. These results demonstrate Chemeleon's unique strength in handling composition-based predictions while maintaining competitive structural accuracy. However, there is still room for improvement in CSP-specific performance, particularly in enhancing structure generation accuracy to match or exceed state-of-the-art CSP models.

**Evaluation of text-guided generation**

Our core objective is to generate compounds based on textual descriptions as conditioning data. The models are assessed with two metrics in terms of conditional generation, which are composition and crystal system matching ratio. These metrics measure the proportion of the 20 sampled structures with the composition or crystal system for the ground truth structure in the test set that was used to generate the textual descriptions. Their detailed scores are provided in Supplementary Table S2.

Figure 3a shows the accumulated composition matching ratio as a function of the number of atoms in unit-cell for the models trained with Baseline BERT and Crystal CLIP as text encoders when the input textual description is in general text form. The Crystal CLIP model significantly outperforms the BERT model across a range of atom number, demonstrating a 3 times higher composition matching ratio. This result highlights the effectiveness of the contrastive learning approach in Crystal CLIP, where text embedding vectors are aligned with graph embeddings that capture the spatial configuration of three-

dimensional structures using equivariant GNNs. However, the composition-matching ratio tends to decline as the number of atoms in the structure increases. It is observed that most mismatches in composition for Crystal CLIP models involve differences in stoichiometric coefficient, rather than in atom types. For instance, when the text input specifies $LiO_3$ with 36 atoms, the model might generate a structure with composition $Li_8O_{26}$ instead of the expected $Li_9O_{27}$. This is due to the nature of the stochastic processes in the generative models. While these models improve the overall distribution under guidance, they cannot always produce the exact conditioning data, particularly when denoising atom types which are categorical variables.

This tendency is further illustrated in the t-SNE plot of compositional embeddings for the training, test, and generated data, as shown in Fig. 3c. The generated data represents sampled structures by Chemeleon when the textual description is general text. These compositional embeddings were generated by averaging element embeddings using Magpie[35], which provides a 22-dimensional representation for each element, through the ElementEmbedding[36] package. Overall, the sampled structures generated by Crystal CLIP mostly exhibit overlap with the structures in the test set, denoted by star markers, suggesting that the models effectively produce consistent compositions. Data points positioned at a significant distance from the test set are highlighted with red edges, indicating that the generated structures have deviated compositions relative to the test set, potentially consisting of different atom types. Notably, a greater number of data points from the Baseline BERT model than from Crystal CLIP fail to overlap with the test set. This suggests that while the Crystal CLIP model maintains a high degree of compositional fidelity, BERT struggles to generate structures that align with the target compositions.

Figure 3d displays the t-SNE plot of the CrystalNN fingerprint[37] for the same structures as in Fig. 3c, which provides a visualization of the structural similarities between the generated and test set structures. The CrystalNN fingerprint, which encodes the local chemical environment of each atom in the crystal structure, is used here to assess how closely the generated structures resemble the ground truth structures at an atomic level. As in the previous plot, data points representing structures that deviate significantly from the test set are highlighted with red edges. BERT demonstrates more deviation from the test structures, indicating that Crystal CLIP is more effective in generating structures that closely match the atomic environments of the ground truth. Several points in the test set, highlighted in Supplementary Fig. S4, are positioned far from the sampled structures. These outliers correspond to structures with complex stoichiometries, such as $Er_2Co_{12}Ni$, $Lu_2Co_{17}$, and $Er_3Ge_{13}Rh_4$, which pose challenges for generation.

Figure 3b presents the composition and crystal system matching ratios of Crystal CLIP models for various types of textual descriptions, including models without textual guidance, specifically for structures containing fewer than 20 atoms. General text descriptions, which are generated by LLMs, exhibit greater flexibility and diversity in linguistic expression. In contrast, composition and formatted text descriptions follow a strict, standardized format with elements listed alphabetically (e.g., $Li(MnO_2)_4$ represented as $Li_1\ Mn_4\ O_8$). Despite the increased complexity and variability of the general text inputs, Crystal CLIP demonstrates robust performance, achieving higher composition and crystal system matching ratios compared to BERT. Notably, the performance gap is most pronounced when using general text descriptions showing the capability of contrastive learning to handle less structured, more naturalistic language inputs.

The strong performance of the general text model in composition and crystal system matching, compared to composition-based and fixed-format inputs, highlights the effectiveness of our approach. Unlike traditional classifier-based or explicitly constrained methods, our methodology enables scalable and flexible material generation,

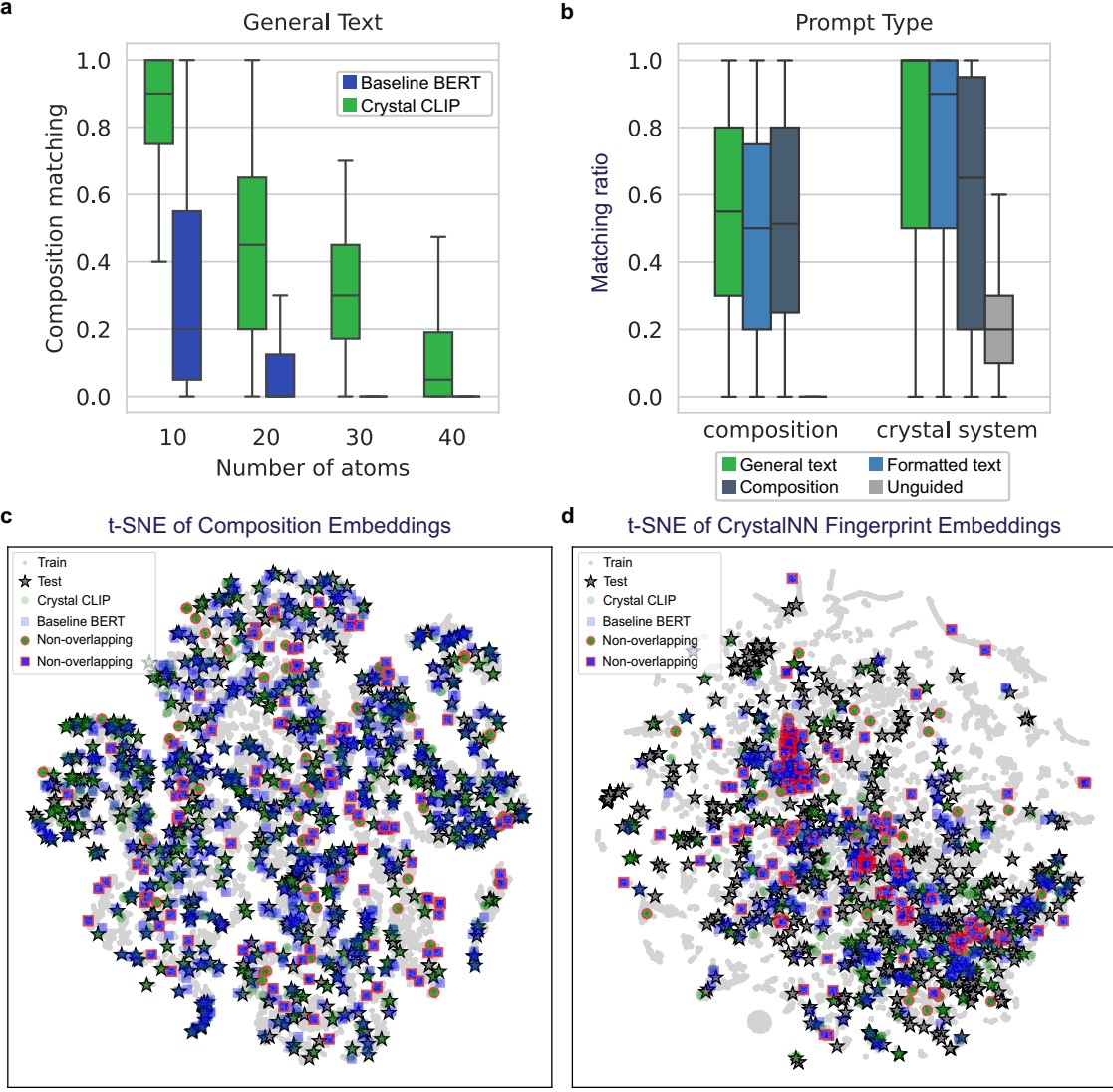

**Fig. 3 | Text-guided compound generation. a** A box plot of the accumulated composition-matching ratio as a function of the number of atoms. The sample sizes ($n$) are: $n = 61$ for structures with fewer than 10 atoms, $n = 300$ for structures with fewer than 20 atoms, $n = 340$ for structures with fewer than 30 atoms, and $n = 707$ for structures with fewer than 40 atoms. Boxes span the 25th to 75th percentiles (the interquartile range), horizontal lines mark the median, whiskers extend to $1.5 \times$ the interquartile range. **b** A box plot of composition and crystal system matching ratio of the Crystal CLIP model based on different prompt types for structures containing fewer than 20 atoms, with a total of 300 generated

structures. The box plot follows the same statistical format and visualization parameters as used in (**a**). **c** A t-SNE plot of compositional embeddings generated by Magpie for train, test, and generated structures from the Baseline BERT and Crystal CLIP models, using general text descriptions. The generated structures that differ from the test set are highlighted with red edges (non-overlapping). **d** A t-SNE plot of structural embeddings generated by CrystalNN Fingerprint, showing train, test, and generated structures using general text descriptions. The source data are provided as a Source Data file 1.

accommodating a broad spectrum of property descriptions. This adaptability suggests a promising avenue for integrating extensive scientific knowledge sources, such as literature and databases, into a unified framework for text-driven material discovery.

### Applications of Chemeleon

To showcase the capabilities of the developed generative AI model, we target several distinct chemical spaces. The first Ti-Zn-O system has been extensively explored and contains many known materials, such as $TiO_2$ and ZnO. The second system is Li-P-S-Cl, representing a complex quaternary space that is sparsely populated, with a limited number of known materials despite its technological relevance in electrochemical energy storage. The challenge of multi-component chemical spaces, like ternary or quaternary systems, lies in their vast size. Even with a maximum stoichiometric coefficient of 6, for instance,

the Li-P-S-Cl system has 2400 possible combinations, calculated as $7^4 - 1$. Each element can have a stoichiometry ranging from 0 to 6 and excluding those where coefficients are zero. This vastness makes comprehensive exploration time-consuming and computationally demanding.

To address this issue, we introduce a tailored workflow integrating multiple computational tools: SMACT[38] (chemical filter), Chemeleon (sampling), MACE-MP (preliminary geometry optimization), and Atomate2[39] (automated density functional theory calculations). At first, the huge search space can be refined to feasible compositions by chemical rules based on electronegativity balance and charge neutrality. In the case of the Ti-Zn-O system, only 179 unique compositions were allowed out of 728 possible combinations, assuming a maximum stoichiometric coefficient of 8. For Li-P-S-Cl, this approach significantly narrowed down the possibilities to 781 unique composition + +ns. The

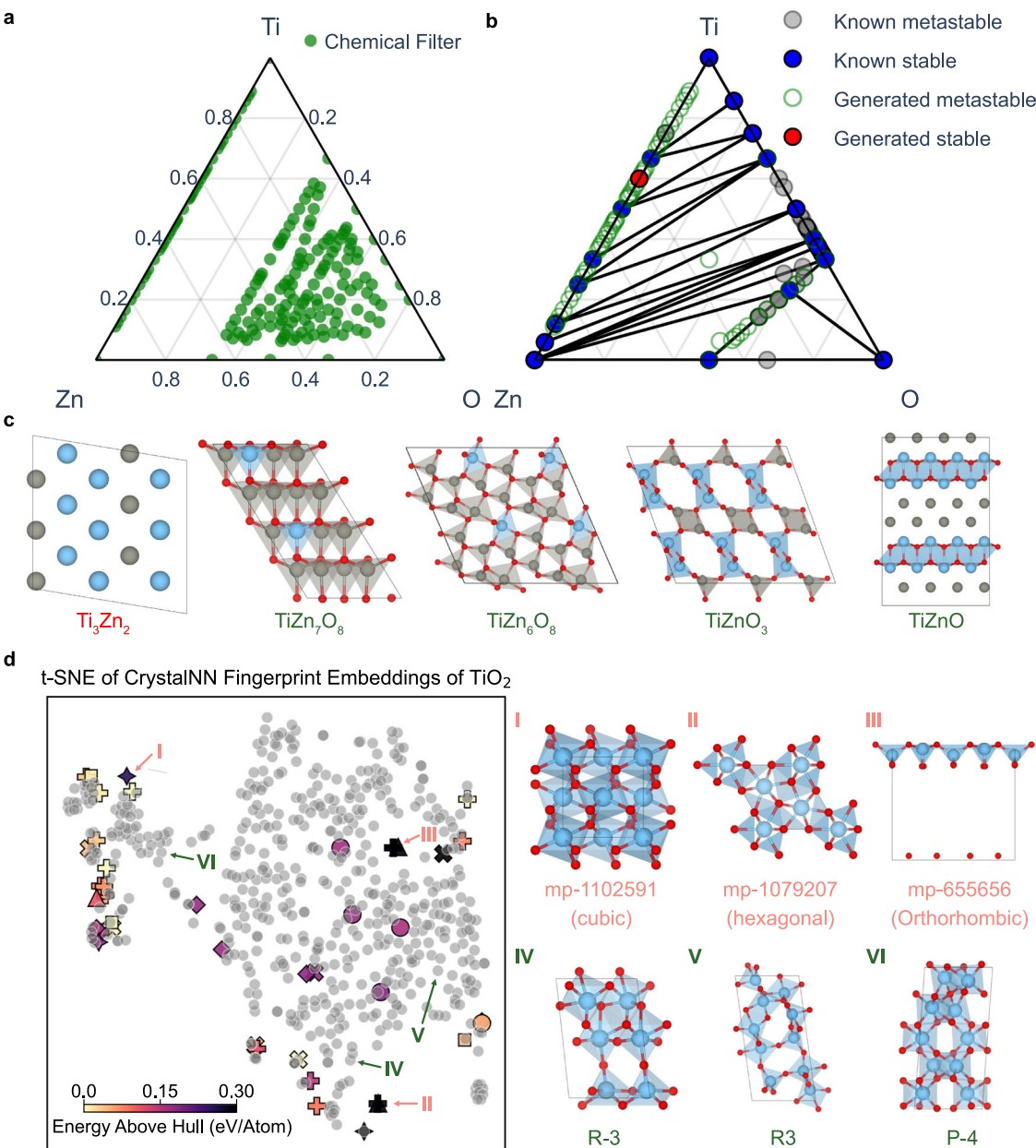

**Fig. 4 | Ternary Zn-Ti-O space. a** Phase diagram with compositions filtered by chemical rules. **b** A combined phase diagram of known and generated compounds by Chemeleon for the Ti-Zn-O system. Stable materials lie on the convex hull, while metastable materials have energies above the convex hull by less than 0.15 eV/atom. **c** Five representative generated structures from the combined phase diagram for Ti-Zn-O system. **d** A t-SNE (t-distributed stochastic neighbor) plot of CrystalNN fingerprint embeddings for TiO₂ polymorphs, including both known and generated structures. It highlights three known TiO₂ polymorphs that were not generated and three generated polymorphs with distinct space groups. The gray dots represent known TiO₂ structures derived from the Materials Project database. The symbols correspond to different crystal systems in generated structures as follows: • (Triclinic), X (Monoclinic), ■ (Tetragonal), + (Orthorhombic), ♦ (Trigonal), ✦ (Cubic), and ▲ (Hexagonal). The source data are provided as a Source Data File 1.

regions of composition allowed by these chemical filters for both the Ti-Zn-O and Li-P-S-Cl spaces are shown in Fig. 4a and Supplementary Fig. S5, respectively.

Subsequently, Chemeleon sampled structures only for compositions that passed the chemical filters, with additional details provided in Supplementary Note S2.

S1. These generated structures were evaluated by calculating the energy above the thermodynamic convex hull relative to phase diagram entries in the Materials Project database. Metastable structures were defined as those with an energy is lower than 0.15 eV/atom of energy above the convex hull. The energy threshold is widely used in

high-throughput computational screening for practical purpose[40] identifying materials with potential synthesizability to account for potential non-equilibrium effects, such as temperature and pressure, by bypassing thermodynamic constraints.

Given the large number of generated structures MACE-MP, was employed for preliminary geometry optimization and energy calculation, identifying potential metastable structures. Finally, the energy calculations for the pre-screened structures were performed using density functional theory (DFT) following geometry optimization. This workflow facilitated the construction of phase diagrams and the

identification of potential novel compounds. The details of workflows for each system are summarized in Supplementary Fig. S6.

## Polymorphs of TiO₂

We first explored the capability to generate specific compositions, focusing on the polymorphs of $TiO_2$. While we do not expect to discover any new stable structures, it is an interesting phase space to explore with a generative model owing to its complexity and importance in solid-state chemistry. The Materials Project database lists 44 $TiO_2$ crystal structures, involving several extensively studied polymorphs. It includes the commonly found rutile, which features an octahedral geometry around each titanium ion in a tetragonal symmetry[41]. Anatase, another well-known tetragonal polymorph close in energy to rutile, but with more distorted octahedra. $TiO_2$ can also exist in other forms, such as brookite and $TiO_2$-B, and these polymorphs can be accessed depending on processing conditions such as temperature, pressure, and chemical environment[42]. This structural flexibility makes $TiO_2$ highly versatile for applications[43].

Using Chemeleon, 549 $TiO_2$ potential polymorphs, distinct from the known structures were sampled with compositions ranging in multiplicity from 1 to 13. Following preliminary geometry optimization using MACE-MP, 539 of these structures successfully converged. To compare the generated polymorphs with the known ones, we constructed a t-SNE plot based on their structural embeddings derived from CrystalNN fingerprints, as shown in Fig. 4d. The names of entries for all known materials are displayed in Supplementary Fig. S7. It is worth highlighting that the generated polymorphs, depicted as light grey points on the plot, not only encompass most known polymorphs but also occupy previously unexplored regions of the structural space.

Certain known polymorphs presented in Fig. 4d, such as structure I (mp-1102591), II (mp-1079207), and III (mp-655656) are positioned in regions not covered by the sampled structures. These were absent from the training set due to their high energy above the convex hull; a filter of 0.25 eV/atom was used to favour the generation of energetically favourable configurations. Indeed, most of the structures generated by Chemeleon exhibit energies below 0.15 eV/atom above the convex hull, according to MACE-MP as shown in Supplementary Fig. S8. The entry mp-1102591 consists of densely packed $TiO_2$ octahedra in a cubic symmetry, while mp-1079207 features sevenfold coordinated Ti with hexagonal symmetry[44]. Lastly, mp-655656 features a square pyramidal coordination of Ti, typically seen in TiO. However, the layers in this structure are spaced far apart, making it difficult for the generative model to sample.

We identified 122 unique metastable $TiO_2$ structures based on the final DFT calculations. Among these, 50 structures exhibit space groups that have not been observed in previously known $TiO_2$ polymorphs, as presented in Supplementary Fig. S9. They differ in terms of local environment, connectivity and/or crystal symmetry. For instance, Structure IV, with space group R-3 in the trigonal system, features a unit cell comprising four octahedra and two square pyramidal units, showcasing a complex coordination environment distinct from typical $TiO_2$ polymorphs. Structure V, with the R3 space group, presents a rare arrangement of both tetrahedral and trigonal bipyramidal coordination polyhedra, highlighting the model's capability to predict uncommon structural motifs. Structure VI is an example of a configuration in the tetragonal system with space group P-4, formed through a unique three-dimensional arrangement of octahedra. Notably, the generative model can uncover plausible new structural configurations in a well-known binary space.

## Extension to Ti-Zn-O

The Ti-Zn-O includes the binary oxides and various zinc titanates such as $Zn_2TiO_4$ and $ZnTiO_3$ with wide-ranging properties[45,46]. Using the developed workflow, the Ti-Zn-O phase diagram in Fig. 4b was constructed by integrating known materials data from the Materials Project with structures generated by Chemeleon. The Materials Project lists 18 stable and 108 metastable structures for this system. Through our workflow, we predicted one stable structure, lying below the energy convex hull, and 58 metastable structures from the generated candidates, where the generated structures are listed in Supplementary Fig. S10.

Figure 4c presents five representative structures generated by Chemeleon. A simple yet effective method for exploring new compounds is through atomic substitution into known prototypes. This strategy is apparent in the metastable structures generated especially for Ti-Zn structures, where several can be viewed as atomic substitutions within known metal structures. The only structure identified below the energy convex hull, $Ti_3Zn_2$, with space group I4/mmm, features alternating layers of Ti and Zn stacked along the c axis. This is an ordered intermetallic structure, which will be in competition with disordered alloys that may form, but are not considered in the standard thermodynamic convex hull analysis. Several other metastable structures were formed by atomic or polyhedral unit substitution. For instance, $TiZn_7O_8$ (space group Cm) is formed with a Zn site in a tetrahedral coordination geometry that is replaced by Ti. While Ti typically prefers octahedral environments in oxides, it can also adopt a tetrahedral geometry, especially in high-pressure scenarios. In the case of $TiZn_6O_8$, the structure appears not to be formed simply through direct atomic substitution but through a unique three-dimensional arrangement incorporating tetrahedral Ti motifs.

The compound $TiZnO_3$, metastable with respect to disproportionation into ZnO and $TiO_2$, combines five-fold coordinated Ti and tetrahedral Zn. This may reflect the training data where Ti can adopt a wide range of coordination environments, while Zn predominantly maintains a +2 oxidation state with a tetrahedral coordination[44]. Structures such as this highlight the potential for generative models to sample unconventional bonding arrangements. Indeed, one sampled metastable structure for TiZnO, which may be challenging for an inorganic chemist to anticipate, separates into alternating layers of TiO and metallic Zn. While the local environments of conventional oxidation states should be well described by our model, we note the potential limitations of the Materials Project training data for highly correlated systems such as those containing reduced forms of Ti.

## Quaternary Li-P-S-Cl system

In the transition from liquid electrolyte to solid-state batteries, the search for solid electrolytes that satisfy all necessary design criteria is ongoing. Argyrodite-type crystals such as $Li_6PS_5Cl$ have emerged as some of the highest-performing systems[47] and the electrode-electrolyte interphase that forms upon Li cycling is crucial for determining performance[48]. At the atomistic scale, this is a challenging problem with a large phase space where phases can form and decay with various stoichiometries under different conditions. Identifying various by-products that form poses a challenge due to the mixture of multiple phases in low concentrations[49]. Understanding such phase behaviour is essential for optimizing battery performance.

As a test for our method, we have tackled the Li-P-S-Cl space where only 16 stable (68 metastable crystals) are present in the Materials Project. Stable crystalline phases are well established for binary compounds at the edges of the phase diagram such as $Li_3P$, LiCl, and $Li_2S$. Using our approach, 17 new stable structures are proposed below the convex hull, along with 435 metastable structures.

Remarkably, only two quaternary entries, $Li_6PS_5Cl$ and $Li_5P(S_2Cl)_2$, are found in the Materials Project. Figure 5a represents the phase diagram of 250 metastable structures, derived from the new convex hull constructed using 435 newly generated quaternary metastable structures. Furthermore, 50 of the quaternary structures with the lowest energy above the convex hull are presented in Supplementary Fig. S11. These findings reveal patterns in the generated

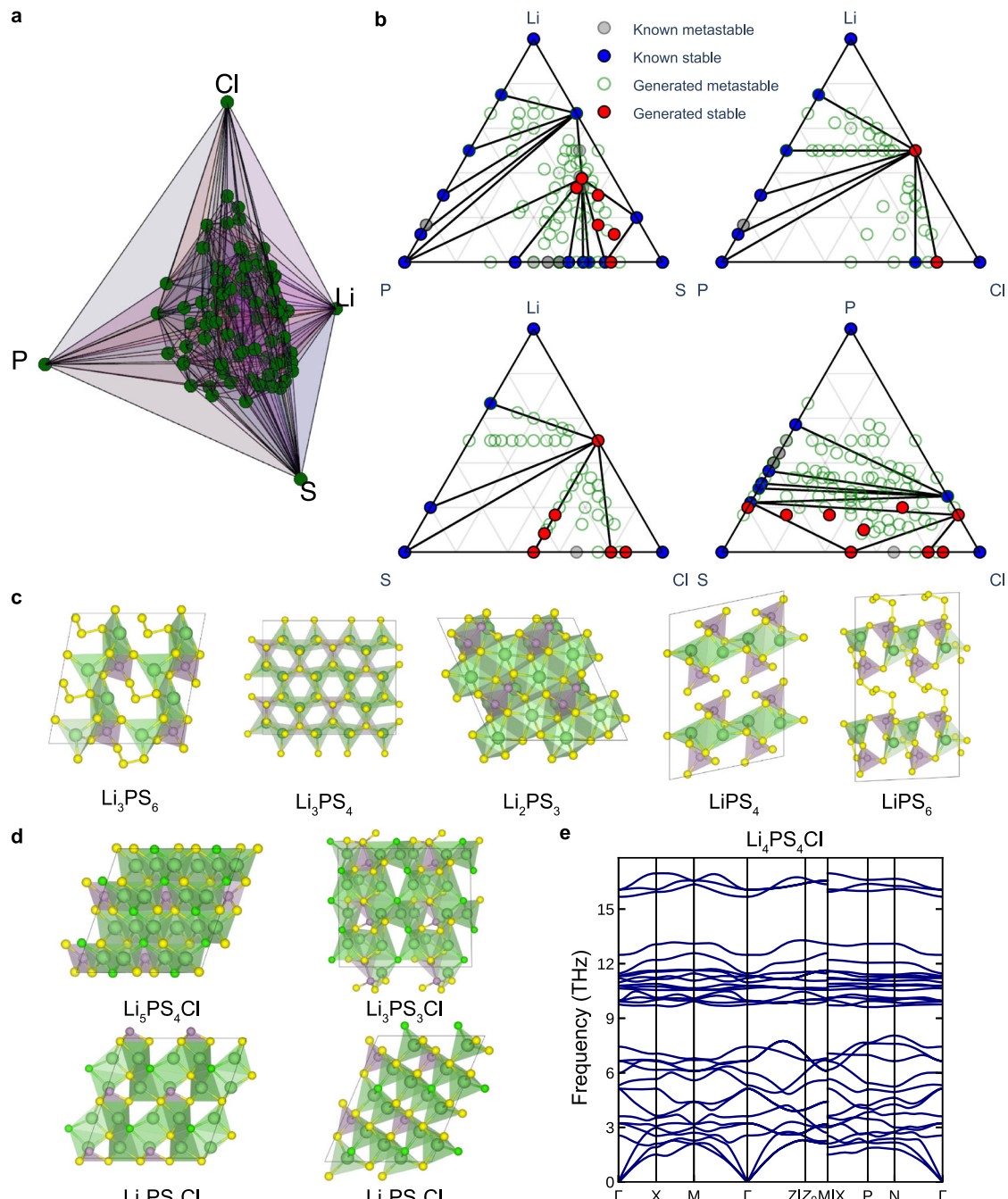

**Fig. 5 | Quaternary Li-P-S-Cl space. a** A phase diagram including generated structures for quaternary materials. **b** Combined phase diagrams of known and generated materials for ternary combinations in Li-P-S-Cl system, including Li-P-S, Li-P-Cl, Li-S-Cl, and P-S-Cl. **c** Five representative generated structures from the combined phase diagram for Li-P-S. **d** Four generated metastable and dynamically stable materials from the combined phase diagram for Li-P-S-Cl (**e**) Phonon dispersion of Li$_4$PS$_4$Cl, exhibiting dynamically stability (absence of imaginary vibrational modes).

structures, particularly with respect to coordination environments. Li frequently exhibits tetrahedral coordination environments involving S or Cl. P tends to form both tetrahedral and trigonal bipyramidal polyhedra with S. The energy distribution of the generated structures, as depicted in Supplementary Fig. S8, reveals that approximately 80% of the sampled configurations have energy values within 0.15 eV/atom above the convex hull. This suggests that Chemeleon has effectively learned stable local coordination environments, even for the Li-S-Cl system that had zero entries in the training set. To further assess the structure stability, vibrational (phonon) calculations were performed.

Figure 5d shows four dynamically stable structures Li$_5$PS$_4$Cl, Li$_6$PS$_4$Cl, Li$_4$PS$_4$Cl, and Li$_3$PS$_3$Cl with their band structures are provided in Fig. 5e and Supplementary Fig. S12. The phonon dispersion for the metastable quaternary structures exhibits no imaginary frequencies throughout the Brillouin zone, indicating that these are true local minima on the potential energy surface.

For binary and ternary systems, the combined phase diagrams with both known and generated structures for ternary systems in Li-P-S-Cl are provided in Fig. 5b. The 17 stable structures generated by Chemelon are presented in Supplementary Fig. S13. Many of these are

molecular crystals, mirroring some of the known phases such as $P_2S_5$ and $PCl_3$ (see Supplementary Fig. S14); however, we note that some contain molecular fragments or voids that appear unlikely to be stable. While $Li_3PS_4$ can exist in multiple crystalline forms including α, β, and γ polymorphs, an alternative stable triclinic configuration on the DFT energy landscape, has been generated as shown in Fig. 5c. In addition, Chemeleon successfully generated structures such as $Li_2PS_3$, $Li_3PS_6$, $LiPS_4$, and $LiPS_6$ with the presence of known thiophosphate motifs characteristic of $(P_2S_6)^{4-}$ and $(PS_4)^{3-}$. Notably, $Li_2PS_3$ also exhibits dynamically stability without imaginary vibrational frequencies as shown in Supplementary Fig. S12. The search of the Li-P-S-Cl space using Chemeleon required approximately 72 h on a single A100 GPU, while a conventional evolutionary algorithm search would have taken orders of magnitude more computational time.

In this workflow, we demonstrate that a text-guided generative model can serve as an efficient and versatile tool for navigating chemical space. In Supplementary Note S3, we conduct an analysis of sampling in practical use cases using various existing CSP methodologies, including diffusion-based DiffCSP, LLM-based CrystaLLM[50], and element substitution[51]. Other methods designed for the CSP task, which primarily focus on optimizing atomic arrangements and lattice matrices based on compositional inputs, can also be integrated into this workflow. These methods, along with our text-guided sampling approach, offer complementary strategies for exploring chemical space.

## Model limitations

While Chemeleon demonstrates considerable potential in navigating chemical systems based on textual descriptions, several limitations must be addressed for broader applicability. There is not a rigid constraint between the input text and the generated compounds, i.e., an input of $Cu_2O_5$ may result in a structure with stoichiometry $Cu_3O_4$ due to the distribution in compositions it is stochastically sampling from. In turn, a request for a cubic crystal may return a unit cell that is not strictly cubic but should be close to the learned conditions $a = b = c$, alpha = beta = gamma = 90°. This issue could be viewed as a feature when it comes to flexible inverse design as the prompt is a text guide rather than hard coded rule or relationship.

One challenge arises when the generation process involves numerical properties, such as band gap. Current text encoders struggle to interpret and generate numerical data accurately. Transformer-based models, while efficient, tend to have difficulty distinguishing between numerical values, especially since many are pre-trained on masked language modelling tasks that aren't designed to handle numbers well. More advanced models, like Transformer decoders or T5 encoders[52] with more sophisticated pre-training, could offer a solution. Moreover, due to the stochastic nature of the generation process, guiding the model to match compositions when many atoms are in the unit cell can be challenging.

To further explore the potential of text-guided generation in Supplementary Note S4, we trained Chemeleon using common mineral types that align with human naming conventions, such as "perovskite" and "spinel". It successfully generates new perovskite-type structures, showcasing the flexibility and scalability of our approach. While our current work focuses on relatively straightforward textual descriptions of composition and crystal systems, future research should expand to more intricate, application-specific scenarios, including constraints on advanced material properties. Moreover, leveraging an expanded range of textual inputs, from abstracts to full research articles, will further enhance the text-guide crystal structure generation's potential from natural language inputs.

In conclusion, we developed a text-guided denoising diffusion model for materials generation. By aligning embeddings from both a text encoder and GNNs through contrastive learning, we improved the model's ability to generate crystal structures guided by textual inputs.

Furthermore, contrastive learning proved more effective when the textual descriptions had diverse styles, rather than relying on strictly formatted text. This approach enabled the successful sampling of crystal structures for well-known systems such as the binary Ti-O and ternary Zn-Ti-O chemical spaces, offering insights into the generative capabilities of AI models. In addition, the model explored less-characterized systems like Li-P-S-Cl, important for solid-state batteries, where both stable and metastable structures were identified, and a revised phase diagram was proposed. While further developments are required to expand towards more complex structures and properties, these methods have already made large-scale computational searches of chemical space more accessible.

## Methods

### Dataset construction

The dataset utilized in this study was curated from the Materials Project database (version 2023.11.01), filtered for unit cells containing 40 atoms or fewer. A total of 32,525 structures were obtained through the Materials Project API, ensuring that all selected structures were experimentally observed and possessing an energy above the convex hull of less than 0.25 eV per atom. Structures with cell lengths exceeding 20 Å were also excluded. Structures that naturally exist as gases at standard room temperature, such as $H_2$, $O_2$, and noble gases, were systematically excluded from the dataset to avoid isolated neutral molecular fragments when generating crystal structures. To rigorously evaluate the model's generalizability to unseen data, a chronological split was applied. Entries registered before August 2, 2018, were allocated to the training and validation set, while those added after this date were used as the test set. Detailed distributions of properties, including the number of entries per year, crystal system, and number of atoms, are provided in Supplementary Fig. S15.

To generate textual descriptions in general text form, the gpt-3.5-turbo model in OpenAI API was used to generate five textual descriptions with diverse styles for each material when its composition and crystal system are given. One of these generated textual descriptions was randomly selected for general text form. The prompt used for generating these descriptions and the examples are detailed in Supplementary Fig. S3.

### Contrastive learning

To achieve alignment between text and graph embeddings within a shared embedding space, we adopt a contrastive learning[24] approach. The text embeddings $E_{text} \in \mathbb{R}^{B \times d}$ are derived from the first token of a BERT model, with an initial dimension of 768, where $B$ is the batch size. The graph embedding $E_{graph} \in \mathbb{R}^{B \times d}$ are obtained by applying average pooling to the out features of the GNN, initially having a dimension of 512. Both embedding vectors are passed through a projection layer that transform them into a unified embedding dimension $d$, 768.

The pairwise similarity matrices for text and graph embeddings are computed by $S_{text} = E_{text}E_{text}^\top$ and $S_{graph} = E_{graph}E_{graph}^\top$, where $S_t$ and $S_g$ are both $\mathbb{R}^{B \times B}$.

The target similarity matrix $T$ is defined as the softmax function of the two similarity matrices:

$$T_{ij} = \frac{\exp\left(\frac{S_{text} + S_{graph}}{2}\right)_{ij}}{\sum_{k=1}^{B} \exp\left(\frac{S_{text} + S_{graph}}{2}\right)_{ik}} \tag{1}$$

which typically approximates an identity matrix where $T \in \mathbb{R}^{B \times B}$.

The logits $R$ representing the similarity between text and graph embeddings, are computed as $R = E_{text}E_{graph}^\top$. The contrastive learning loss $\mathscr{L}_{contrastive}$ is computed by cross-entropy when the logits and target distributions. The text-to-graph loss $\mathscr{L}_t$ and graph-to-text loss

$\mathscr{L}_g$ are defined as:

$$\mathscr{L}_{text} = -\frac{1}{B}\sum_i^B\sum_j^B T_{ij}\log\left(\frac{\exp(R_{ij})}{\sum_{k=1}^B\exp(R_{ik})}\right) \quad (2)$$

$$\mathscr{L}_{graph} = -\frac{1}{B}\sum_i^B\sum_j^B T_{ij}\log\left(\frac{\exp\left(R_{ji}\right)}{\sum_{k=1}^B\exp\left(R_{jk}\right)}\right) \quad (3)$$

The final loss $\mathscr{L}_{contrastive}$ is the average of these two components:

$$\mathscr{L}_{contrastive} = \frac{1}{2}(\mathscr{L}_{graph} + \mathscr{L}_{text}), \quad (4)$$

The training of contrastive learning was implemented with a batch size of 128, using the Adam optimizer. The learning rate for graph and text encoders are set to $10^{-4}$ and $10^{-5}$, respectively. The training was conducted over 1000 epochs with early stopping, which is triggered if the validation loss doesn't improve for 300 epochs. A learning rate scheduler with a reduction on plateau mechanism was employed, with a patience parameter set to 200 epochs.

## Diffusion model

A denoising diffusion model is employed for generating crystal structures which are represented as $C = (A, X, L)$, where $A \in \mathbb{Z}^N$ represent the atom types, $X \in \mathbb{Z}^{N \times 3}$ denote the fractional coordinates of the atoms within the unit cell, and $L \in \mathbb{Z}^{3 \times 3}$ is the lattice matrix defining the unit cell dimensions and angles. The diffusion process comprises a forward process, which the stochastic degradation (or corruption) of the crystal structures, while the reverse process iteratively learns the pattern to gradually reconstruct the data distribution from pure noise. The forward process is $q(C_{1:T}|C_0) = \prod_{t=1}^T q(C_t|C_{t-1})$, which add noise to the original structure $C_0$ over $T$ time steps, while the reverse process $p_\theta(C_{0:T}) = p(C_T)\prod_{t=1}^T p_\theta(C_{t-1}|C_t)$ is a Markov process that iteratively denoises the corrupted structure, reconstructing the data distribution from pure noise to sample crystal structures. The generative models for crystal structures consists of three distinct diffusion processes, each designed to a specific component of the crystal structures: atom type $A$, atomic fractional coordinates $X$, and lattice matrix $L$.

## Diffusion process for atom types, $A$

Atom types are modelled as discrete categorical variables, where each atom type is represented as a discrete label from a finite set. Denoising Diffusion Models in Discrete State-Spaces (D3PM)[53] approach was adopted, which adapts the diffusion process to suit discrete state spaces. In the forward process, it gradually corrupts the atom types over time by introducing noise through a series of transition matrices, while typically adding random noise such as Gaussian noise during corruption. Specifically, an absorbing transition matrix $Q_t$ at each time step $t$ transforms the original atom types into noise state. The absorbing state ensures that the distribution $A_t$ becomes independent of the initial atom types $A_0$ as $t$ becomes sufficiently large. The corrupted atom types at time $t$, $A_t$ are sampled by:

$$q(A_t|A_0) = Cat(A_t; p = A_0\bar{Q}_t), \quad (5)$$

where $Cat(x; p)$ represents a categorical probability distribution with parameter $p$, and $\bar{Q}_t = Q_1 Q_2 \dots Q_t$ is the cumulative product of the transition matrices $Q_t$ at each time step. $T$, The absorbing transition matrices in the D3PM approach is adopted, where it is designed to transform $A_T$ into dummy atom, of which atomic number is zero, at the final time step.

The training loss function for this process is derived from the negative variational lower bound (VLB), augmented by an auxiliary cross-entropy loss term. The VLB is expressed as:

$$\mathcal{L}_{vb} = E_{q(A_0)}\big[D_{KL}\big[(q(A_t|A_0))||p(A_T)\big]$$
$$+ \sum_{t=2}^T \mathbb{E}_{q(A_t|A_0)}\big[D_{KL}\big[q(A_{t-1}|A_t,A_0)||P_\theta(A_{t-1}|A_t)\big]\big] \quad (6)$$
$$- \mathbb{E}_{q(A_1|A_0)}[\log p_\theta(A_0|A_1)]\big],$$

where each term represents loss for $t = T, t - 1, 0$, respectively. To further enhance the model's ability to accurately reconstruct the atom types, an auxiliary cross-entropy loss term $L_{ce}$ is added, which directly penalizes the discrepancy between the predicted atom types and the true atom types at time step $t = 0$. Therefore, the final loss is defined by:

$$\mathscr{L}_A = \mathscr{L}_{vb} + \lambda_{ce}\mathbb{E}_{q(A_0)}\mathbb{E}_{q(A_t|A_0)}[-\log p_\theta(A_0|A_t)], \quad (7)$$

where $\lambda_{ce}$ is a coefficient that controls the weight of the cross-entropy loss, which is set to 1.0 in this work.

## Diffusion process for lattice matrix, $L$

The lattice matrix $L$ is a continuous variable representing the unit cell parameters, consisting of $3 \times 3$ matrix. To model the diffusion process of lattice matrix, Diffusion Probabilistic Model[28] framework was adopted. In the forward process, Gaussian noise is added to the lattice matrix at each time step $t$, preserving the variance of the original distribution. Given an initial lattice matrix $L_0$, the noisy lattice matrix in forward process is sampled by:

$$q(L_t|L_0) = N(L_t|\sqrt{\bar{\alpha}_t}L_0, (1 - \bar{\alpha}_t)I), \quad (8)$$

Where $\bar{\alpha}_t = \prod_{s=1}^t \alpha_t = \prod_{s=1}^t(1 - \beta_s)$ is the cumulative product of noise schedule parameters where $\beta_t \in (0, 1)$. Therefore, the noisy lattice at time step $t$, is computed as:

$$L_t = \sqrt{\bar{\alpha}_t}L_0 + \sqrt{1 - \bar{\alpha}_t}\epsilon_L, \quad (9)$$

where $\epsilon_L \sim N(0, I)$ is Gaussian noise. The reverse process is parameterized by $\theta$, predicting the distribution $p_\theta(L_{t-1}|L_t)$ to reconstruct $L_0$. The loss function for the lattice parameters is the mean squared error (MSE) between the true noise added to the lattice and the predicted noise:

$$\mathscr{L}_{L,t} = \mathbb{E}_{q(L_t|L_0)}(||\epsilon_L - \hat{\epsilon}_\theta(L_t, t)||^2), \quad (10)$$

The loss is computed as the average across random time steps $t$, uniformly sampled from the interval $[0, T]$.

## Diffusion process for atom coordinate, $X$

The fractional coordinates $X$ of atoms within the unit cells are continuous variables bounded within the domain $[0, (1)^3$. Modeling the diffusion process for these coordinates presents unique challenges due to their cyclical and bounded nature. To address this, we employ a Score-Matching (SM) based framework[54,55], which is a variance-exploding diffusion process. It which has been demonstrated as effective for generative models for fractional coordinate. The forward process was carried out with the Wrapped Normal (WN) distribution[56], $N_w$ and the variance $\sigma_t^2$ at time t:

$$q(X_t|X_0) = N_w(X_t|X_0, \sigma_t^2 I), \quad (11)$$

This noise is wrapped around the domain $[0, (1)^3$ to ensure that the atomic position remain within bounded region while preserving the periodicity.

The noisy data for fractional coordinate $X_t$ at time step t is computed as:

$$X_t = w(X_0 + \sigma_t \epsilon_X) \qquad (,12)$$

where $\epsilon_X$ is sample from a standard normal distribution $N(0, I)$ and $w(\cdot)$ is the truncation function that ensures the coordinates remain within the domain by wrapping them around the boundaries. The training objective is to minimize MSE between the actual noise $\epsilon_X$ and the predicted noise $\hat{\epsilon}_\theta$:

$$\mathscr{L}_{X,t} = \mathbb{E}_{q(X_t|X_0)}(||\epsilon_X - \hat{\epsilon}_\theta(X_t, t)||^2) \qquad (,13)$$

The loss is averaged over random time steps $t$ uniformly sampled from the range $[0, T)$.

## Training details

The overall loss is a weighted sum of the losses for atom types, lattice matrix, and atom coordinate:

$$\mathscr{L}_{total} = \lambda_A \mathscr{L}_A + \lambda_L \mathscr{L}_L + \lambda_X \mathscr{L}_X \qquad (,14)$$

$\lambda_A$, $\lambda_L$, $\lambda_X$ represents weights for atom type, lattice matrix, and atom coordinate, respectively, with each weight set to 1.0. Model training was performed using a single A100 80GB GPU with the same batch size and learning rate scheduler in the contrastive learning, except for the learning rate $10^{-3}$ for diffusion models.

## Text-guided generation

The text-guided generation is implemented through classifier-free guidance[30], which uses text embeddings as condition data. In classifier-free guidance, the model is trained to conditionally generate samples based on both the input text and an unconditional distribution, allowing it to flexibly adjust the level of guidance based on the text input. Given text embeddings $\mathbf{E}_{text} \in \mathbb{R}^{B \times d}$ obtained from the pretrained Crystal CLIP, the conditional distribution at each time step can be represented as:

$$p_\theta(A_{t-1} L_{t-1} X_{t-1} | A_t L_t X_t E_{text,t}), \qquad (15)$$

To balance learning from both conditional and unconditional distributions, the model is trained with a conditional dropout strategy, where the conditioning text input is randomly omitted with a probability of 0.2.

During inference, the classifier-free guidance is applied by interpolating between the conditional and unconditional predictions. This is expressed by modifying the predicted noise as:

$$\widetilde{\epsilon}_\theta(X_t, E_{text,t} = (1 - \gamma)\epsilon_\theta(X_t) + \gamma \epsilon_\theta(X_t, E_{text,t}), \qquad (16)$$

where $\gamma$ is a weighting factor that determines the extent of text guidance influence. In this work, $\gamma$ is set to 2.0, which biases the generation process towards the text guidance.

To effectively incorporate text embeddings as conditioning data, Feature-wise Linear Modulation (FiLM)[57] layers are employed. FiLM layers modulate the intermediate feature representations of the neural network by applying affine transformations, where the scaling and shifting parameters are functions of the text embeddings. This mechanism enables the network to adjust its internal feature activations dynamically in response to the input text, thereby aligning the generated outputs more closely with the intended textual guidance.

## Denoising network

We adopted the equivariant GNN with respect to $E(3)$ transformations, including translation, rotation, and reflection developed by DiffCSP[14]. It is built on the EGNN[58] framework, which achieves geometric equivariance in a computationally efficient manner by explicitly encoding Cartesian coordinate information during the message-passing process. To further enhance the model's ability to handle crystal structures with periodic invariance along with the $E(3)$ symmetries, they introduced a Fourier transformation on the fractional coordinates of the atoms. This transformation plays a crucial role in facilitating periodic E(3)-invariant outputs by capturing the inherent periodicity in the lattice structures of crystals. The use of fractional coordinates, compared to Cartesian coordinates, is particularly powerful when combined with the WN diffusion process. This approach aligns with the intrinsic periodicity of crystal lattices, allowing for more accurate modelling of atomic positions.

## First-principles calculations

All DFT calculations carried out in this work were performed using the Vienna ab initio simulation package (VASP)[59,60] within the Projector Augmented Wave formalism[61] using Atomate2[39], FireWorks[62], Pymatgen, and Custodian[32]. For consistency with the Materials Project to build phase diagrams of the generated structures, the relaxation and static classes used the MPGGADoubleRelaxStaticMaker class to create the workflows for each structure, including using the PBE exchange-correlation functional[63]. For phonon calculations, the PhononMaker class in atomate2.vasp.flows.phonons module was used. For constructing supercells, a minimum length of 10 Å is used instead of the default of 20 Å to reduce the computational cost. The phonon computations were started from the primitive standardised structures using the PBEsol functional[64], which provides a better description of unit cell volumes and phonon frequencies of crystals.

## Reporting summary

Further information on research design is available in the Nature Portfolio Reporting Summary linked to this article.

## Data availability

The MP-40 dataset utilized in this study are available at the Zenodo repository: https://doi.org/10.5281/zenodo.15090949. The DFT-optimised structures in TiO2, Ti-Zn-O, Li-P-S-Cl system generated by Chemeleon, are provided in Supplementary Data (raw CIF files). Source data are provided with this paper.

## Code availability

The source code for Chemeleon is available at the following GitHub repository: https://github.com/hspark1212/chemeleon/. To ensure reproducibility of this study, the specific version of Chemeleon (v0.1.1) is archived at Zenodo: https://doi.org/10.5281/zenodo.15090949. The training and testing logs are available on Weights & Biases at: https://wandb.ai/hspark1212/Chemeleon_v0.1.1.

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

## Acknowledgements

We thank Youhan Lee and Junkil Park for insightful discussion. This work was supported by EPSRC project EP/X037754/1. Via our membership of the UK's HEC Materials Chemistry Consortium, which is funded by EPSRC (EP/X035859/1), this work used the ARCHER2 UK National Supercomputing Service (http://www.archer2.ac.uk). We are grateful to the UK Materials and Molecular Modelling Hub for computational resources, which is partially funded by EPSRC (EP/T022213/1, EP/W032260/1 and EP/P020194/1). We acknowledge computational resources and support provided by the Imperial College Research Computing Service (https://doi.org/10.14469/hpc/2232). The project also benefited from membership of the AI for Chemistry: AIchemy hub (EPSRC grant EP/Y028775/1 and EP/Y028759/1).

## Author contributions

H.P. and A.W. conceptualized this project. The model building, training and testing were performed by H.P. The DFT workflow was set up and run by A.O. All authors contributed to the data analysis and writing of the manuscript.

## Competing interests

The authors declare no competing interests.
