## [Transparent Peer Review file · Nature Communications]

Exploration of crystal chemical space using text-guided generative artificial intelligence

Corresponding Author: Professor Aron Walsh

Version 0:

Reviewer comments:

Reviewer #1

(Remarks to the Author)

The manuscript by Park et al. presents an innovative contribution to the field of crystal structure generation by using advanced ML techniques such as graph neural networks (GNNs) and cross-modal learning to explore structural and chemical spaces. The framework presented is well-suited for high-throughput exploration, offering a scalable tool for identifying promising candidates in materials science. The manuscript is organized and presented clearly, making it accessible to readers from diverse backgrounds in computational materials science and machine learning. The work combines state-of-the-art techniques to deliver valuable insights that are likely to inspire further research in the field, so I am happy to recommend publication in Nature Communications. However, I recommend the authors address the following points before the manuscript is accepted:

1. In the Introduction, the authors state that “materials exploration has evolved from the study of individual systems to high-throughput screening”. I think such statements are not useful as they oversimplify the state of the field: while high-throughput methods have indeed gained prominence, in-depth studies of individual systems remain critically important for understanding materials, refining methods, and validating predictions. In fact, high-throughput studies still rely heavily on rough approximations, and they are generally valid only to extract trends. The authors should revise their statement to acknowledge the complementary roles of in-breadth (high-throughput) vs in-depth (focused) studies, perhaps emphasizing how they collectively advance the field.

2. I understand the motivation for the chronological splitting of the test set, to simulate a realistic scenario of predicting “future” compounds. However, I wonder if that could potentially introduce some data leakage. Newer Materials Project entries will often include derivatives or light variations of earlier structures, such as doped versions or polytypes. Is it possible that it is therefore easier to predict such new structures than a randomly selected set of unseen structures from the whole database? This could result in over-optimistic performance metrics. I suggest the authors discuss in more detail whether their chronological split adequately tests generalization to truly novel structures.

3. I am surprised by the prediction of stability of Ti_3Zn_2 (in terms of being lower in energy than the convex hull), given the well-established stability of $TiZn$ and Ti_2Zn in the Ti-Zn phase diagram. Ti_3Zn_2 is not reported in experimental phase diagrams, and the thermodynamic dominance of $TiZn$ and Ti_2Zn (see Okamoto, Ti-Zn (titanium-zinc). *Journal of Phase Equilibria and Diffusion*, 29, 211-212, 2008) suggests that any combination of these two phases should be more stable than Ti_3Zn_2 . I suggest that the authors explicitly compare the formation energy of Ti_3Zn_2 against that of $TiZn$ and Ti_2Zn mixtures, investigate potential errors in convex hull construction or evaluation, and explain what might stabilize Ti_3Zn_2 if their prediction holds, perhaps with a discussion of possible reasons why this phase has not been observed experimentally so far.

4. The use of the generalised gradient approximation, GGA (PBE functional) for DFT calculations in the Ti-Zn-O space, raises concerns about accuracy, especially when describing Ti compounds with oxidation states below +4. GGA is known to poorly handle localized d-electrons, leading to underestimated band gaps and wrong formation energies. For example, Ti_2O_3 is predicted to be a metallic compound by GGA, while it is actually a narrow-gap semiconductor. That means that the GGA phase diagram of titanium oxides can be expected to be severely distorted in comparison with experiment. While there

are well-established DFT corrections that been shown to significantly improve predictions for transition metal oxides, they are not applied here, presumably to align with Materials Project settings. The authors should justify this choice, acknowledging and discussing the potential impact on phase stability predictions. Can their results in the Ti-Zn-O system be trusted at all?

5. The manuscript uses a threshold of 0.15 eV/atom above the convex hull to define metastability, which is perhaps a bit too permissive. Energy differences between polymorphs are typically much smaller (e.g., 10–50 meV/atom), and materials exceeding 0.1 eV/atom above the hull are rarely experimentally realizable. The authors should justify this threshold.

6. The manuscript would benefit from some benchmarking that puts this work in context with recent advances in the field of ML-generated crystal structures. There is a range of new methods out there, including not only diffusion-based methods but also LLM-based ones. Comparing performance and functionality with these other models would be useful for the reader and the community to understand the strengths and limitations of Chemeleon.

(Remarks on code availability)

Reviewer #2

(Remarks to the Author)

The manuscript by Park and colleagues introduces a generative model for crystalline materials, designed to facilitate efficient exploration of the vast chemical space for material design. The authors employ textual inputs aligned with structural information via cross-modal contrastive learning to guide a diffusion model within the framework of classifier-free guidance. This novel approach aims to enhance the generation performance of crystalline structures by leveraging the synergy between textual and structural data.

The performance of the proposed generative model was evaluated using proxy metrics such as validity, uniqueness, match rate, and metastability. The results demonstrate that diffusion models guided by text embeddings through contrastive learning achieve a significantly improved match rate compared to those guided by text embeddings without contrastive learning. To further validate their model, the authors applied it to the exploration of ternary Ti-Zn-O and quaternary Li-P-S-Cl systems, showcasing its ability to efficiently navigate complex material systems.

While I appreciate the innovative ideas underlying the proposed model, the primary criterion for evaluating any method should be the rigor and quality of its performance assessment. In this regard, I find that the manuscript falls short of providing a comprehensive or robust evaluation of the generative model's effectiveness. Below, I outline my reasons for this assessment and provide my suggestions for improvement.

1. The authors state that their core objective is to "generate compounds based on textual descriptions as conditioning data." To achieve this, they utilize text-based data containing information about the chemical composition and crystal system as conditioning inputs to guide the diffusion model. However, I believe that using basic structure-based information, such as chemical composition or specific symmetry, as conditioning inputs may not be the most suitable choice for achieving the stated objective. As far as I am aware, several existing diffusion models, such as MatterGen, DiffCSP, and DiffCSP++, have already demonstrated the capability to perform this type of conditional generation in a more straightforward and effective manner. For instance, in MatterGen, chemical composition or space group information is directly encoded as a conditional input for the diffusion model, which is trained using classifier-free guidance. Similarly, in DiffCSP, the chemical composition can be simply fixed within the diffusion model. These models have achieved state-of-the-art performance across various metrics, including those utilized by the authors in this study.

To substantiate the efficacy of the proposed method, I suggest that the authors either explore alternative conditioning properties beyond basic structural information—such as conductivity or other material properties—or provide a direct comparison with existing methods, such as DiffCSP or FlowMM. This would better contextualize the contribution of their approach and clarify its advantages relative to established benchmarks.

2. Regarding the metrics listed in Table I, I recommend that the authors include a comparison with existing models that have performed similar evaluations on standard datasets such as MP20 and MPTS52. For instance, FlowMM achieves a match rate of 61% on MP20 and 17% on MPTS52. Including such comparisons would provide a clearer context for evaluating the performance of the proposed method and help demonstrate its relative strengths and weaknesses against state-of-the-art approaches.

3. Again, for the two applications involving the Ti-Zn-O and Li-P-S-Cl systems, while the integration of the proposed model into a high-throughput workflow has led to the discovery of new stable or metastable compounds, the results do not clearly demonstrate the superiority of the current method over existing approaches. It remains unclear how the performance of the workflow would compare if the current model were replaced with alternative structure generation methods, such as crystal structure prediction techniques, prototype substitution, or other generative models.

In addition, there are some minor revisions that need to be addressed to improve the clarity and quality of the manuscript:

1. In the caption of Figure 5(d), the "Li-P-S" appears to be incorrect and should likely be "Li-P-S-Cl."

2. The authors state that the validity metric is defined by the absence of atomic overlap. However, it is not clear to what extent atomic overlap is considered or how it is quantified.

In conclusion, the authors have proposed a promising generative model for crystal structures. However, the results presented in the study do not fully demonstrate the superiority of the proposed model compared to existing approaches. A more comprehensive comparison with established methods is necessary to better highlight the advantages and limitations of the model.

(Remarks on code availability)

Reviewer #3

(Remarks to the Author)

The manuscript by Park et al describe the development of a generative model for crystal structures for materials discovery. Generative models such as these, particularly diffusion models, have been explored extensively in prior instances such as CDVAE, DiffCSP, MatterGen and many others. The primary innovation of the manuscript involves the conditioning of the generation on textual input, rather than on the more commonly encountered materials properties. This conditioning is accomplished using CLIP, while the generation is based on an existing method, DiffCSP. However, it does not appear that this feature enables generation of structures which is otherwise not possible with existing methods; i.e. prior studies such as MatterGen are equally capable of generating specific space groups or symmetries even without the textual conditioning. As no examples were provided to suggest the textual conditioning can go beyond simple space group and symmetries to more complex constraints, the current manuscript is not particularly compelling or suitable for publication in Nature Communications.

Additional comments:

-A more compelling demonstration of text-guided generation needs to show examples of generation which *cannot* be otherwise accomplished by other existing methods. For example, including additional descriptions or constraints, such as vacancies/defects, dopants, specific coordination environments, disorder, etc. Otherwise, it is not clear what the method brings to the table.

-Furthermore, the demonstrative examples should be chosen/set-up in such a way to showcase the uniqueness or advantages of text-guided generation; as it currently stands, generation in the Ti/O and Li/P/S/Cl space is very generic and can be easily accomplished with a number of different approaches, from generative to more traditional CSP (non-ML). What makes Li/P/S/Cl well-suited for text-guided generation, and how can text-guided generation enable the discovery of new phases not possible with some other baseline(s)?

(Remarks on code availability)

Version 1:

Reviewer comments:

Reviewer #1

(Remarks to the Author)

The authors have reasonably addressed all the comments and requests by the reviewers. I am happy to recommend publication of the manuscript in its current form.

(Remarks on code availability)

Reviewer #2

(Remarks to the Author)

The proposed Chameleon model presents a novel and impactful approach to crystal structure generation by effectively integrating textual descriptions with structural data. I believe the focus should be on the innovative idea behind the proposed model, rather than an extensive comparison with existing methods that involve incremental technical optimizations. The authors have successfully addressed all my concerns; therefore, I recommend the publication of this work in Nature Communications.

(Remarks on code availability)

Reviewer #3

(Remarks to the Author)

The authors have written a response to the points raised in the review, but somewhat disappointingly did not make any attempts to address them in the methodology or results. The main concern still stands, namely, this is a minimally viable demonstration of text-guided generation, but does not provide any real advantages or additional capabilities compared to non text-guided approaches. This is most evident in the demonstration on the Li/P/S/Cl system - which can be explored using any existing generative method besides this one. The authors themselves cannot provide a convincing use case where this method is uniquely well-suited. As such the case for publication remains low.

(Remarks on code availability)

Point by point response

Reviewer # 1

The manuscript by Park et al. presents an innovative contribution to the field of crystal structure generation by using advanced ML techniques such as graph neural networks (GNNs) and cross-modal learning to explore structural and chemical spaces. The framework presented is well-suited for high-throughput exploration, offering a scalable tool for identifying promising candidates in materials science. The manuscript is organized and presented clearly, making it accessible to readers from diverse backgrounds in computational materials science and machine learning. The work combines state-of-the-art techniques to deliver valuable insights that are likely to inspire further research in the field, so *I am happy to recommend publication in Nature Communications*. However, I recommend the authors address the following points before the manuscript is accepted:

1. In the Introduction, the authors state that “materials exploration has evolved from the study of individual systems to high-throughput screening”. I think such statements are not useful as they oversimplify the state of the field: while high-throughput methods have indeed gained prominence, in-depth studies of individual systems remain critically important for understanding materials, refining methods, and validating predictions. In fact, high-throughput studies still rely heavily on rough approximations, and they are generally valid only to extract trends. The authors should revise their statement to acknowledge the complementary roles of in-breadth (high-throughput) vs in-depth (focused) studies, perhaps emphasizing how they collectively advance the field.

(Response)

We thank the reviewer for the positive recommendation and for highlighting the importance of both in-depth and high-throughput screening and their complementary roles for accelerating materials discovery. In the revised manuscript, we have clarified the complementary roles of these approaches in advancing the field.

(Changes made)

In Introduction on Page 2, we have revised the statement as follows:

“Computational materials exploration has expanded to incorporate both in-depth studies of individual systems¹ and high-throughput screening² approaches, each playing a complementary role in advancing the field. While high-throughput methods have significantly broadened the scope of materials discovery³, in-depth studies remain essential for uncovering fundamental mechanisms and validating computational predictions. However, as these search spaces grow, so does the complexity of identifying regions of interest where chemical compositions, crystal structures, and physical properties align to create materials with desirable characteristics.”

2. I understand the motivation for the chronological splitting of the test set, to simulate a realistic scenario of predicting “future” compounds. However, I wonder if that could potentially introduce some data leakage. Newer Materials Project entries will often include derivatives or light variations of earlier structures, such as doped Materials Project entries will often include derivatives or light variations of earlier structures, such as doped versions or polytypes. Is it possible that it is therefore easier to predict such new structures than a randomly selected set of unseen structures from the whole database? This could result in over-optimistic performance metrics. I suggest the authors discuss in more detail whether their chronological split adequately tests generalization to truly novel structures.

(Response)

We acknowledge the concern that newer entries in the Materials Project database may often be derived from previously existing materials, such as doped or polytypes. While the time-based (chronological) split has been used in generative AI owing to providing insights into the predictive performance on future, unseen materials, it could lead to over-optimistic performance metrics due to potential similar structures from the training set.

To further investigate this, we trained Chameleon with the composition and general text-type prompts, after training the contrastive learning (Crystal CLIP as text encoders), using a random split of the MP-40 dataset. Interestingly, compared to the original time-based split as shown in Table 1 and Supplementary Table 3, structure matching, metastability, and composition matching exhibited an improvement of over

two times, although uniqueness decreased. Additionally, metastability and crystal system matching (in the general-text prompt) also showed significant improvements.

These results suggest that the complexity of newer MP entries may influence model performance. While the test set in the time-based split primarily consisted of structures with unit cells containing 18 or more atoms, the random split produced a more balanced distribution of atom counts across the dataset, ranging from 2 to 40 atoms per unit cell. This underscores the impact of dataset splitting strategies on model evaluation and offers valuable insights into performance variations.

(Changes made)

We have included the model evaluation results using a random split in Supplementary Table S3 and the atom distribution comparison in Supplementary Figure S16.

Supplementary Table S3. Model evaluation on structural and guiding metrics with random split. Evaluation of the crystal structures generated by Chemeleon in terms of validity (structural parameters), uniqueness (between generated structures), structure matching (ground truth present), and metastability (energy threshold), composition matching and crystal system (lattice) matching across a batch of 20 samples.

Model type	Text encoder	Validity	Uniqueness	Structure matching	Metastability	Composition matching	Crystal system matching
composition	Crystal CLIP	0.99	0.72	0.41	0.37	0.50 (0.66)	0.61 (0.65)
general text	Crystal CLIP	0.99	0.72	0.43	0.36	0.50 (0.66)	0.84 (0.86)

(): Values in parentheses represent scores for structures having fewer than 20 atoms

Supplementary Figure S16. Number of atoms in MP-40 test set with different data splits. Distributions of number of atoms in MP-40 test set with different splits: time-based split (blue) and random split (orange).

We have also added the following discussion in Supplementary Note S1, with an additional sentence in the "Evaluation of Structure Metrics" section on Page 9:

"Detailed information about the dataset is provided in the Methods section and a comparison between chronological and random splits is discussed in Supplementary Note S1."

"Supplementary Note S1. Model evaluation on structural and guiding metrics with random split.

To ensure a robust evaluation across different dataset splitting methods, we also assessed the performance of Chameleon trained with a random split on the MP-40 dataset, as shown in Supplementary Table S3. This random split exhibit improved performance in structure matching, metastability, composition matching, and crystal system matching compared to the time-based split. This improvement can be attributed to the increased complexity of newer entries added to the Materials Project after August 2, 2018. As shown in Supplementary Figure S16, the distribution of the number of atoms per unit cell varies significantly between the two splitting methods. While the test set in the time-based split predominantly consists of structures with more atoms, the test set in the random split yields a more balanced distribution of atom counts.”

3. I am surprised by the prediction of stability of Ti_3Zn_2 (in terms of being lower in energy than the convex hull), given the well-established stability of TiZn and Ti_2Zn in the Ti-Zn phase diagram. Ti_3Zn_2 is not reported in experimental phase diagrams, and the thermodynamic dominance of TiZn and Ti_2Zn (see Okamoto, Ti-Zn (titaniumzinc). Journal of Phase Equilibria and Diffusion, 29, 211-212, 2008) suggests that any combination of these two phases should be more stable than Ti_3Zn_2 . I suggest that the authors explicitly compare the formation energy of Ti_3Zn_2 against that of TiZn and Ti_2Zn mixtures, investigate potential errors in convex hull construction or evaluation, and explain what might stabilize Ti_3Zn_2 if their prediction holds, perhaps with a discussion of possible reasons why this phase has not been observed experimentally so far.

(Response)

This is a fair point and indeed we had not intended to identify any new configurations on that line of the phase diagram. While confirmed that the structure is a valid local minimum, there is an implicit assumption of ordered crystal energies in these phase diagrams and there is no competition with solid solutions whose free energy will be lowered due to configurational entropy. This is not an issue of the generative model, but with the standard approximation for computing such phase diagrams.

(Changes made)

We have clarified this point in the text on Page 18 of the main text to emphasise the potential limitations:

“This is an ordered intermetallic structure, which will be in competition with disordered alloys that may form, but are not considered in the standard convex hull analysis.”

4. The use of the generalised gradient approximation, GGA (PBE functional) for DFT calculations in the Ti-Zn-O space, raises concerns about accuracy, especially when describing Ti compounds with oxidation states below +4. GGA is known to poorly handle localized d-electrons, leading to underestimated band gaps and wrong formation energies. For example, Ti_2O_3 is predicted to be a metallic compound by GGA, while it is actually a narrow-gap semiconductor. That means that the GGA phase diagram of titanium oxides can be expected to be severely distorted in comparison with experiment. While there are well-established DFT corrections that have been shown to significantly improve predictions for transition metal oxides, they are not applied here, presumably to align with Materials Project settings. The authors should justify this choice, acknowledging and discussing the potential impact on phase stability predictions. Can their results in the Ti-Zn-O system be trusted at all?

(Response)

We agree and this is a widespread approximation that will need to be overcome in the future. The largest curated datasets are based on GGA calculations, and most generative models and indeed machine learning force fields are trained and tested on calculations based on Materials Project. This avoids the need to recompute all competing phases in the relevant phase diagram. One of the case studies in the recent Mattering paper in Nature is Sr-V-O, which is a chemical space known for its high correlation. The main justification is that crystal structures are better represented than the associated electronic structures, with a range of Ti(IV) and Ti(III) environments being present in the Materials Project.

(Changes made)

We have added a statement on the potential limitations on Page 18:

“While the local environments of conventional oxidation states should be well described by our model, we note the potential limitations of the Materials Project training data for highly correlated systems such as the reduced forms of Ti.”

5. The manuscript uses a threshold of 0.15 eV/atom above the convex hull to define metastability, which is perhaps a bit too permissive. Energy differences between polymorphs are typically much smaller (e.g., 10–50 meV/atom), and materials exceeding 0.1 eV/atom above the hull are rarely experimentally realizable. The authors should justify this threshold.

(Response)

We thank the reviewer’s comment regarding the chosen metastability threshold. While many experimentally observed materials are found within 10–50 meV/atom above the convex hull, slightly larger cutoffs, such as 0.15 eV/atom, are frequently adopted in high-throughput computational studies. In the metastability analysis paper by Sun et al.⁴, 50.5% of known compounds are reproduced with a metastability threshold of 10 meV. This is a compromise between recovering true positive cases of known materials and also predicting false positive stable compounds. We have now included additional clarifications and references in the revised manuscript to further justify this choice.

(Changes made)

In “Applications of Chemeleon” section (Page 16) we have added the following:

“Metastable structures were defined as those with an energy is lower than 0.15 eV/atom of energy above the convex hull. The energy threshold is widely used in high-throughput computational screening for practical purpose⁴ identifying materials with potential synthesizability to account for potential non-equilibrium effects, such as temperature and pressure, by bypassing thermodynamic constraints”

6. The manuscript would benefit from some benchmarking that puts this work in context with recent advances in the field of ML-generated crystal structures. There is a range of new methods out there, including not only diffusion-based methods but also LLM-based ones. Comparing performance and functionality with these other models would be useful for the reader and the community to understand the strengths and limitations of Chemeleon.

(Response)

We fully agree with benchmarking to compare performance with other models, leading to better understanding of Chemeleon for practical use cases. To address this, we have conducted benchmarking for three compositions: ZnO, TiO₂, and TiZnO₃ within the Zn-Ti-O system. We used DiffCSP as a diffusion-based method, and CrystaLLM as a LLM-based approach. Additionally, a traditional prototype substitution method is included, an element substitution method based on machine learning developed by Kusaba et al.⁵ For this benchmarking, we sampled 100 structures using each of these different methods and analysed their performance in terms of composition match, validity, uniqueness, and thermodynamic stability (using machine learning forcefield, MACE-MP used in our work)

(Changes made)

We have included the benchmarking results in Supplementary Figure S17 and S18 and analysis in Supplementary Note S3.

“Supplementary Note S3. Benchmarking Chemeleon with other methods.

Beyond evaluating Chemeleon using the MP-40 test set, we benchmarked its performance against other structure generation methods to assess for practical use cases. This benchmarking study focuses on three compositions within the Zn-Ti-O system: ZnO, TiO₂, and TiZnO₃, considering different integer variations of TiZnO₃ such as Ti₂Zn₂O₆ and Ti₄Zn₄O₁₂. We compared Chemeleon with three other methods: DiffCSP, a diffusion-based crystal structure prediction (CSP) model; CrystaLLM, a large language model (LLM)-based approach; and an element substitution method based on machine learning, developed by Kusaba et al.⁵

For this study, we sampled 100 structures for each composition using each method and evaluated their performance based on composition match, validity, uniqueness, and thermodynamic stability, as shown in Supplementary Figure S17 and S18. The thermodynamic stability of the generated structures was assessed using the MACE-MP-0 machine-learning force field, which we utilized throughout our study.

In terms of composition matching, DiffCSP and element substitution methods are developed for CSP task, without additional optimisation of composition prediction. They inherently achieve a perfect composition matching score of 1.0. For Chemeleon, we used composition-based model of which inputs are string format of composition. While it shows almost perfect composition accuracy for relatively simple compositions such as ZnO and TiO₂, the composition matching performance declines as stoichiometry increases and the unit cell size becomes larger. For instance, In the case of Ti₄Zn₄O₁₂, some generated structures deviated to Ti₃Zn₅O₁₂ or Ti₄Zn₃O₁₃. In contrast, CrystaLLM exhibited strong composition generation capabilities.

The validity of generated structures was evaluated based on structural abnormalities, including atomic overlap or excessively large unit cell parameters (greater than 60 Å). All methods predominantly generated valid structures, with a few invalid structures observed in CrystaLLM for Ti₂Zn₂O₆.

To evaluate uniqueness, we analysed the proportion of unique structures using `StructureMatcher` class from pymatgen. The element substitution method exhibits the highest uniqueness compared to other generative AI methods. This is due to the nature of substitution methods, which generate structures from diverse structural prototypes, leading to a broader range of unique crystal structures.

Supplementary Figure S18 shows that distribution of the energy per atom based on MACE-MP-0 model. The predicted structures by the substitution method are very broad, indicating that most of the generated structures have higher energy and may be less stable. CrystaLLM exhibits an energy distribution peak in a relatively unstable region compared to DiffCSP and Chemeleon. Furthermore, the energy distribution of CrystaLLM-generated structures is more sparsely spread out, rather than concentrated at lower energy values.

Overall, DiffCSP, which is optimized exclusively for CSP rather than composition prediction, generates structures with the lowest energy distribution peak, indicating superior performance of generating stable structures. Chemeleon, which is optimized to generate crystal structures from textual input, produces structures with an energy distribution comparable to DiffCSP, while also demonstrating flexibility in handling composition variations.”

Supplementary Figure S17. Benchmarking Chameleon with other methods. The composition matching, validity, uniqueness, when sampling 100 structures for ZnO, TiO₂, TiZnO₃, considering different integer variations of TiZnO₃ such as Ti₂Zn₂O₆ and Ti₄Zn₄O₁₂. Three other methods are assessed including DiffCSP, a diffusion-based crystal structure prediction (CSP) model; CrystaLLM, a large language model (LLM)-based approach; and an element substitution method based on machine learning.

Supplementary Figure S18. Energy distribution in generated crystal structures in benchmarking results. (a) Energy per atom distribution of sampled structures obtained from the MACE-MP-0 model when sampling 100 structure for ZnO, TiO₂, TiZnO₃, considering different integer variations of TiZnO₃ such as Ti₂Zn₂O₆ and Ti₄Zn₄O₁₂. Three other methods are assessed including DiffCSP, a diffusion-based crystal structure prediction (CSP) model; CrystaLLM, a large language model (LLM)-based approach; and an element substitution method (b) Same data as in (a) but with a reduced y-axis range for improved visualization.

Reviewer #2 (Remarks to the Author):

The manuscript by Park and colleagues introduces a generative model for crystalline materials, designed to facilitate efficient exploration of the vast chemical space for material design. The authors employ textual inputs aligned with structural information via cross-modal contrastive learning to guide a diffusion model within the framework of classifier-free guidance. This novel approach aims to enhance the generation performance of crystalline structures by leveraging the synergy between textual and structural data.

The performance of the proposed generative model was evaluated using proxy metrics such as validity, uniqueness, match rate, and metastability. The results demonstrate that diffusion models guided by text embeddings through contrastive learning achieve a significantly improved match rate compared to those guided by text embeddings without contrastive learning. To further validate their model, the authors applied it to the exploration of ternary Ti-Zn-O and quaternary Li-P-S-Cl systems, showcasing its ability to efficiently navigate complex material systems. While I appreciate the innovative ideas underlying the proposed model, the primary criterion for evaluating any method should be the rigor and quality of its performance assessment. In this regard, I find that the manuscript falls short of providing a comprehensive or robust evaluation of the generative model's effectiveness. Below, I outline my reasons for this assessment and provide my suggestions for improvement.

1. The authors state that their core objective is to "generate compounds based on textual descriptions as conditioning data." To achieve this, they utilize text-based data containing information about the chemical composition and crystal system as conditioning inputs to guide the diffusion model. However, I believe that using basic structure-based information, such as chemical composition or specific symmetry, as conditioning inputs may not be the most suitable choice for achieving the stated objective. As far as I am aware, several existing diffusion models, such as MatterGen, DiffCSP, and DiffCSP++, have already demonstrated the capability to perform this type of conditional generation in a more straightforward and effective manner. For instance, in MatterGen, chemical composition or space group information is directly encoded as a conditional input for the diffusion model, which is trained using classifier-free guidance. Similarly, in DiffCSP, the chemical composition can be simply fixed within the diffusion model. These models have achieved state-of-the-art performance across various metrics, including those utilized by the authors in this study. To substantiate the efficacy of the proposed method, I suggest that the authors either explore alternative conditioning properties beyond basic structural information—such as conductivity or other material properties—or provide a direct comparison with existing methods, such as DiffCSP or FlowMM. This would better contextualize the contribution of their approach and clarify its advantages relative to established benchmarks.

(Response)

We thank their reviewer for their careful assessment. We acknowledge that employing classifier-free (or classifier-based) guidance or explicitly encoding properties as constraints in the diffusion process are effective techniques for property-guided generation. However, these methods necessitate the development and compilation of separate models (and labelled training data) for each specific target property. In contrast, our approach integrates textual descriptions that can contain a wider range of information that will be easier to generalise. Our findings indicate that general text prompts, despite their inherent complexity, achieve slightly improved performance in composition and crystal system matching when compared to more structured, formatted text inputs. This will ultimately enable models that are easier to train, use and scale, e.g. using extensive knowledge repositories such as scientific literature or textbooks.

While our goal was less a competition between techniques, but more about introducing an alternative approach, we agree that comparing different models is useful. To address this, we have incorporated additional analysis in our responses and revised manuscript:

- Reviewer 1's Comment 6: Presents a comparison of chameleon with other methodologies including DiffCSP, CrystaLLM, element substitution on specific compositions.
- Reviewer 2's Comment 2: Includes results from our model trained on the MP-20 dataset, allowing for direct performance evaluation.
- Reviewer 3's Comment 1: Provides additional analysis using Chameleon, specifically trained on mineral types.

(Changes made)

To enhance the clarity of contributions and advantages in our approach, we have incorporated the following statement into the revised manuscript (Page 13):

“The strong performance of the general text model in composition and crystal system matching, compared to composition-based and fixed-format inputs, highlights the effectiveness of our approach. Unlike traditional classifier-based or explicitly constrained methods, our methodology enables scalable and flexible material generation, accommodating a broad spectrum of property descriptions. This adaptability suggests a promising avenue for integrating extensive scientific knowledge sources, such as literature and databases, into a unified framework for text-driven material discovery.”

2. Regarding the metrics listed in Table I, I recommend that the authors include a comparison with existing models that have performed similar evaluations on standard datasets such as MP20 and MPTS52. For instance, FlowMM achieves a match rate of 61% on MP20 and 17% on MPTS52. Including such comparisons would provide a clearer context for evaluating the performance of the proposed method and help demonstrate its relative strengths and weaknesses against state-of-the-art approaches.

(Response)

We agree and to address this, we have now trained Chameleon on the MP-20 dataset using a composition-based text prompt as input and compared its Structure Matching performance when sampling 20 structures. Chameleon achieved a structure matching rate of 36.14 %, which is lower than DiffCSP’s 77.93% in the crystal structure prediction (CSP) task.

It is important to note that CSP models such as DiffCSP and FlowMM focus solely on optimizing atomic coordinates and lattice matrices to generate stable and plausible crystal structures. In contrast, Chameleon is trained to generate crystal structures directly from text prompts, making a direct comparison with CSP-focused models less appropriate. Specifically, among the 20 sampled structures, some structures exhibited variations in composition from ground true structures, which inherently reduces the structure matching rate. This result highlights both the challenges and opportunities associated with Chameleon. While there is a need to enhance its CSP performance, the results also demonstrate its capability to generate plausible crystal structures solely from textual descriptions.

(Changes made)

We have included the following statement in “Evaluation of structure metrics” section (Page 10) and the results of performance evaluation on MP-20 dataset in Supplementary Table S1.

“Although our model is not solely focused on the CSP task but rather emphasizes alignment for text-guided generation, we also trained it on the MP-20 dataset to facilitate a direct comparison with existing CSP models (see Supplementary Table S1). Unlike models only for the CSP task, which focus solely on optimizing atomic coordinates and lattice matrices, Chameleon incorporates composition as a text-based input. As a result, it achieves a 67.52% composition matching rate, a metric that DiffCSP and FlowMM do not account for.

For structure match rate, our evaluation considers only cases where composition is correctly predicted. This stricter criterion naturally results in a lower structure match rate for Chameleon compared to DiffCSP and FlowMM, as these models do not impose composition constraints. In terms of root mean square error (RMSE), Chameleon underperforms DiffCSP and exhibits similar performance with FlowMM. These results demonstrate Chameleon’s unique strength in handling composition-based predictions while maintaining competitive structural accuracy. However, there is still room for improvement in CSP-specific performance, particularly in enhancing structure generation accuracy to match or exceed state-of-the-art CSP models.”

“Supplementary Table S1. Model evaluation on MP-20 dataset. Comparison of crystal structures generated by Chameleon, DiffCSP, and FlowMM on the MP-20 dataset when sampling 20 structures. The composition match rate and structure match rate indicate the proportion of sampled structures that match the ground truth compositions and structures in the test set. RMSE represents the root mean square error between the sampled structures and the ground truth structures, as implemented in the DiffCSP paper.”

Model type	# of samples	Composition Match Rate	Structure Match Rate	RMSE
FlowMM	20	-	76.55	0.0834
DiffCSP	20	-	77.93	0.0492
Chameleon	20	67.52	36.14	0.0804

3. Again, for the two applications involving the Ti-Zn-O and Li-P-S-Cl systems, while the integration of the proposed model into a high-throughput workflow has led to the discovery of new stable or metastable compounds, the results do not clearly demonstrate the superiority of the current method over existing approaches. It remains unclear how the performance of the workflow would compare if the current model were replaced with alternative structure generation methods, such as crystal structure prediction techniques, prototype substitution, or other generative models.

(Response)

We acknowledge that a comparison with standard crystal structure generation methods would better provide a better understanding of workflow. We have evaluated our approach against traditional prototype substitution and two other generative models, DiffCSP and CrystaLLM. As presented in Supplementary Note S3, our results suggest that while element substitution is computationally efficient and capable of identifying a few low-energy structures, diffusion-based methods, such as Chameleon and DiffCSP, are more effective in generating a diverse range of metastable structures.

The primary objective of this workflow was to demonstrate the feasibility of using a text-guided generative model for chemical space exploration rather than claiming outright superiority over existing approaches. Our findings indicate that text-guided crystal structure generation can be an efficient tool for navigating chemical space. While our current implementation is not yet optimized for the CSP tasks, it highlights the potential of leveraging text-based generative models for structural exploration. As future work, we aim to expand our method by incorporating a broader and more diverse set of text-based inputs and training data, which we anticipate will further enhance its predictive power and versatility in crystal structure generation.

(Changes made)

To further clarify in the proposed workflow, we have added the following sentences to the "Quaternary Li-P-S-Cl System" Discussion section (on Page 21) in the revised manuscript:

"In this workflow, we demonstrate that a text-guided generative model can serve as an efficient and versatile tool for navigating chemical space. In Supplementary Note S3, we conduct an in-depth analysis of sampling in practical use cases using various existing CSP methodologies, including diffusion-based DiffCSP, LLM-based CrystaLLM, and element substitution. Other methods designed for the CSP task, which primarily focus on optimizing atomic arrangements and lattice matrices based on compositional inputs, can also be integrated into this workflow. These methods, along with our approach, offer complementary strategies for exploring chemical space."

4. In the caption of Figure 5(d), the "Li-P-S" appears to be incorrect and should likely be "Li-P-S-Cl."

(Response)

We thank the reviewer for catching this typo error. We have corrected the caption accordingly in the revised manuscript.

(Changes made)

In Figure 5(d) caption, we have revised the sentence to

"Four generated metastable and dynamically stable materials from the combined phase diagram for Li-P-S-Cl"

5. The authors state that the validity metric is defined by the absence of atomic overlap. However, it is not clear to what extent atomic overlap is considered or how it is quantified.

(Response)

We appreciate the reviewer's request for clarification on the definition and quantification of atomic overlap. To address this, we have added an explanation in the manuscript.

(Changes made)

In the "Evaluation of Structure Metrics" section (Page 9), we have added the following sentences:

"The absence of atomic overlap is evaluated by measuring the minimum pairwise interatomic distance. Structures with a minimum pairwise distance greater than 0.5 Å are considered free of atomic overlap."

In conclusion, the authors have proposed a promising generative model for crystal structures. However, the results presented in the study do not fully demonstrate the superiority of the proposed model compared to existing approaches. A more comprehensive comparison with established methods is necessary to better highlight the advantages and limitations of the model.

Reviewer #3 (Remarks to the Author):

The manuscript by Park et al describe the development of a generative model for crystal structures for materials discovery. Generative models such as these, particularly diffusion models, have been explored extensively in prior instances such as CDVAE, DiffCSP, MatterGen and many others. The primary innovation of the manuscript involves the conditioning of the generation on textual input, rather than on the more commonly encountered materials properties. This conditioning is accomplished using CLIP, while the generation is based on an existing method, DiffCSP. However, it does not appear that this feature enables generation of structures which is otherwise not possible with existing methods; i.e. prior studies such as MatterGen are equally capable of generating specific space groups or symmetries even without the textual conditioning. As no examples were provided to suggest the textual conditioning can go beyond simple space group and symmetries to more complex constraints, the current manuscript is not particularly compelling or suitable for publication in Nature Communications.

(Response)

We thank the reviewer for their time and feedback. We recognise the rapid advancements in diffusion-based models for crystal structure generation, which can generally be categorised into two main approaches:

(1) **Crystal structure prediction (CSP)**: aim to generate energetically favourable (meta)stable structures given compositional input,

(2) **De novo generation**: classifier-free guidance for generation (without predefined composition) to sample materials with specific target properties.

A third emerging approach is **text-guided crystal structure generation**, which holds immense potential for leveraging the vast repository of scientific literature and its rich qualitative information. Similar advancements have been observed in other domains, such as text-to-image⁶, as well as in scientific fields like small molecule⁷, protein⁸, where text-based research is rapidly being explored.

We acknowledge that previous methods can guide structures toward specific compositions or crystal systems using classifier-free guidance. *However, our key contribution is demonstrating that **textual input itself** can serve as an extensible and flexible interface for crystal structure generation.* Just as the CLIP⁹ model plays a foundational component of text-to-image generative models, our Crystal CLIP approach lays the groundwork for text-to-crystal structure generation, a rapidly evolving field with significant potential for materials discovery.

1. A more compelling demonstration of text-guided generation needs to show examples of generation which **cannot** be otherwise accomplished by other existing methods. For example, including additional descriptions or constraints, such as vacancies/defects, dopants, specific coordination environments, disorder, etc. Otherwise, it is not clear what the method brings to the table.

(Response)

We thank the reviewer's insightful feedback and fully agree that showcasing examples of text-guided generation beyond what existing methods can accomplish would strengthen our approach. Indeed, incorporating intricate details such as vacancies, dopants, specific local coordination environments, or disorder is an exciting direction that we plan to explore. Text-guided generation offers the flexibility to encode these more complex constraints directly into prompts, without requiring carefully curated or labelled data for each new property.

We fully agree that further developments will be needed to realise its full potential. Our next steps involve refining the model for more complex property constraints with enhanced guiding performance, which are those highlighted by the reviewer, so that text-guided generation can ultimately produce novel materials guided by natural language inputs with precisely engineered characteristics.

In our work, we focused on a foundational demonstration to illustrate the promise of text-based control in crystal structure generation. As an illustrative example, we trained the Chameleon model using intuitive **mineral names**, aligned with common human naming conventions, to generate novel perovskite structures. This proof-of-concept, though relatively simple, demonstrates how our approach is extensible and versatile framework, suggesting that it can accommodate more diverse and sophisticated textual description in future iterations.

(Changes made)

We have included the statement in “Model Limitations” section on page 22 by reflecting the reviewer’s comment.

“To further explore the potential of text-guided generation in Supplementary Note S4, we trained Chameleon using mineral types that align with human naming conventions, such as “perovskite” and “spinel”. It successfully generates new perovskite-type structures, showcasing the flexibility and scalability of our approach. While our current work focuses on relatively straightforward textual descriptions of composition and crystal systems, future research should expand to more intricate, application-specific scenarios, including constraints on advanced material properties. Moreover, leveraging an expanded range of textual inputs, from abstracts to full research articles, will further enhance the text-guide crystal structure generation’s potential from natural language inputs.”

We have included the mineral name results in Supplementary Figure S19 and analysis in Supplementary Note S4.

“Supplementary Note S4. Chameleon trained with mineral types

As a proof-of-concept for the effectiveness of text-guided generation, we trained the Chameleon model using human-recognized mineral name conventions (e.g., perovskite, Heusler, ilmenite). We collected mineral names via Robocrystallography based on the AFLOW prototype database for the MP-40 dataset, including only those with at least 10 occurrences. This resulted in a total of 6,537 structures. The most common mineral names included “(Cubic) Perovskite” and “Orthorhombic Perovskite,” as well as other well-known names such as Spinel and Rocksalt (see Supplementary Figure S19(a)). We randomly split this dataset into 6,000 training structures and 537 test structures.

For evaluation, we sampled 20 structures from the 537 test structures using their mineral name as text inputs and applied the same structure-matching metrics used previously. This resulted in a 0.86 mineral-matching rate.

To further showcase perovskite generation, we sampled 100 structures covering all unique atomic numbers in the training set with “Perovskite” as text inputs. From these, we identified 112 newly generated perovskite structures that did not overlap with the original training or test sets. Among these, three representative examples are shown in Supplementary Figure S19(b). These range from the simple ABX_3 -type perovskite (e.g., $YFeO_3$, formed via element substitution) to more complex quaternary and quinary perovskites. This result demonstrates the model’s capability to generate novel materials based on intuitive, human-recognized text inputs.”

Supplementary Figure S19. Chameleon trained with mineral names. (a) Mineral name distributions were obtained using RoboCrystallographer with the AFLOW prototype database on a training set of 6,000 entries. (b) Three generated Perovskite structures, which are not overlapped in training and test set. The samples are generated using Chameleon trained with mineral names.

- Furthermore, the demonstrative examples should be chosen/set-up in such a way to showcase the uniqueness or advantages of text-guided generation; as it currently stands, generation in the Ti/O and Li/P/S/Cl space is very generic and can be easily accomplished with a number of different approaches, from generative to more traditional CSP (non-ML). What makes Li/P/S/Cl well-suited for text-guided generation, and how can text-guided generation enable the discovery of new phases not possible with some other baseline(s)?

(Response)

We do agree the reviewer's perspective on the importance of demonstrative examples that highlight the unique strengths of text-guided generation. As noted in our response to Reviewer 2's Comment 3, our focus was to demonstrate that the text-guided generative model can effectively explore chemical space while integrating into our workflow.

Our choice of Ti/O and Ti/Zn/O systems was intended to validate the effectiveness of our workflow in well-explored chemical spaces. In contrast, while understanding of Li-P-S-Cl system is crucial in battery systems, conventional approaches such as evolutionary algorithms remain unexplored in this domain. This system presents a significant challenge due to its wide range of stoichiometries and structural complexity. Additionally, the presence of larger unit cells increases the computational cost of intermediate DFT validations, further compounding the difficulty of efficient exploration. In general,

multi-component solids remain a major challenge in materials design, making them particularly well-suited for text-guided generative approaches.

(Changes made)

We have added the following statement in the “Quaternary Li-P-S-Cl System” section on Page 21:

“The search of the Li-P-S-Cl space using Chameleon required approximately 72 hours on a single A100 GPU, while a conventional evolutionary algorithm search would have taken orders of magnitude more computational time”

Reference

1. Heitler, W. & London, F. Wechselwirkung neutraler Atome und homöopolare Bindung nach der Quantenmechanik. *Z. Physik* **44**, 455–472 (1927).
2. Curtarolo, S. *et al.* The high-throughput highway to computational materials design. *Nature Materials* **12**, 191–201 (2013).
3. Butler, K. T., Davies, D. W., Cartwright, H., Isayev, O. & Walsh, A. Machine learning for molecular and materials science. *Nature* **559**, 547–555 (2018).
4. Sun, W. *et al.* The thermodynamic scale of inorganic crystalline metastability. *Science Advances* **2**, e1600225 (2016).
5. Kusaba, M., Liu, C. & Yoshida, R. Crystal structure prediction with machine learning-based element substitution. *Computational Materials Science* **211**, 111496 (2022).
6. Rombach, R., Blattmann, A., Lorenz, D., Esser, P. & Ommer, B. High-Resolution Image Synthesis with Latent Diffusion Models. Preprint at <https://doi.org/10.48550/arXiv.2112.10752> (2022).
7. Liu, S. *et al.* Multi-modal molecule structure–text model for text-based retrieval and editing. *Nat Mach Intell* **5**, 1447–1457 (2023).
8. Ferruz, N., Schmidt, S. & Höcker, B. ProtGPT2 is a deep unsupervised language model for protein design. *Nat Commun* **13**, 4348 (2022).
9. Radford, A. *et al.* Learning Transferable Visual Models From Natural Language Supervision. Preprint at <https://doi.org/10.48550/arXiv.2103.00020> (2021).

AUTHOR'S RESPONSE TO REVIEWERS

Firstly, we thank Reviewer 1 and 2 for their positive response to our changes and for recommending acceptance.

In the first round of revisions, we made a series of changes including model retraining and testing to meet Reviewer #3s suggestions. For their remaining concern, regarding the clarity of our framework's advantages and applications, we emphasize that our approach showcases the generation of crystal structures guided by diverse textual prompts. This framework which is already competitive to other approaches can be further enhanced by integrating intuitive domain-specific data, such as mineral names, into the text-driven generation process.

Over the next few years, we are committed to enhancing our framework by integrating more extensive and specialized textual datasets derived from scientific literature and textbooks. This will be a significant undertaking that will broaden the model's capability to generate based on increasingly complex and precise descriptions including defects and more complex processes as suggested by the Reviewer.